# The OncoPPi network of cancer-focused protein–protein interactions to inform biological insights and therapeutic strategies

Zenggang Li[1,*], Andrei A. Ivanov[1,*], Rina Su[1,2,3,*], Valentina Gonzalez-Pecchi[1], Qi Qi[1], Songlin Liu[1,2], Philip Webber[1], Elizabeth McMillan[4], Lauren Rusnak[1], Cau Pham[1], Xiaoqian Chen[1,5], Xiulei Mo[1], Brian Revennaugh[1], Wei Zhou[6,7], Adam Marcus[6,7], Sahar Harati[8], Xiang Chen[2], Margaret A. Johns[1], Michael A. White[4], Carlos S. Moreno[7,8,9], Lee A.D. Cooper[7,8,10], Yuhong Du[1,7], Fadlo R. Khuri[6,7,†] & Haian Fu[1,6,7]

As genomics advances reveal the cancer gene landscape, a daunting task is to understand how these genes contribute to dysregulated oncogenic pathways. Integration of cancer genes into networks offers opportunities to reveal protein–protein interactions (PPIs) with functional and therapeutic significance. Here, we report the generation of a cancer-focused PPI network, termed OncoPPi, and identification of >260 cancer-associated PPIs not in other large-scale interactomes. PPI hubs reveal new regulatory mechanisms for cancer genes like *MYC*, *STK11*, *RASSF1* and *CDK4*. As example, the NSD3 (WHSC1L1)–MYC interaction suggests a new mechanism for NSD3/BRD4 chromatin complex regulation of MYC-driven tumours. Association of undruggable tumour suppressors with drug targets informs therapeutic options. Based on OncoPPi-derived STK11-CDK4 connectivity, we observe enhanced sensitivity of STK11-silenced lung cancer cells to the FDA-approved CDK4 inhibitor palbociclib. OncoPPi is a focused PPI resource that links cancer genes into a signalling network for discovery of PPI targets and network-implicated tumour vulnerabilities for therapeutic interrogation.

[1] Department of Pharmacology and Emory Chemical Biology Discovery Center, Emory University, Atlanta, Georgia 30322, USA. [2] Department of Dermatology, Xiangya Hospital, Central South University, Changsha 410008, China. [3] Department of Dermatology, Beijing Chao-yang Hospital, Capital Medical University, Beijing 100020, China. [4] Department of Cell Biology, University of Texas Southwestern Medical Center, Dallas, Texas 75220, USA. [5] Department of Pathophysiology, School of Basic Medicine, Huazhong University of Science and Technology, Wuhan 430030, China. [6] Department of Hematology & Medical Oncology, Emory University, Atlanta, Georgia 30322, USA. [7] Winship Cancer Institute, Emory University, Atlanta, Georgia 30322, USA. [8] Department of Biomedical Informatics, Emory University School of Medicine, Atlanta, Georgia 30322, USA. [9] Department of Pathology and Laboratory Medicine, Emory University School of Medicine, Atlanta, Georgia 30322, USA. [10] Department of Biomedical Engineering, Georgia Institute of Technology/Emory University School of Medicine, Atlanta, Georgia 30322, USA. * These authors contributed equally to this work. † Present address: American University of Beirut, Riyad El-Solh, Beirut 11072020, Lebanon. Correspondence and requests for materials should be addressed to F.R.K. (email: frkhuri@aub.edu.lb and fkhuri@emory.edu) or to H.F. (email: hfu@emory.edu).

Protein–protein interactions (PPIs) form the backbone of signal transduction pathways and networks in diverse physiological processes[1]. Due to their critical roles in relaying cell growth signals in both normal and cancer cells, once 'undruggable' PPIs have attracted much attention as a potential new class of drug targets[2,3]. In support of this pursuit, large-scale proteomics approaches have been utilized to generate highly informative PPI interactomes[4–6]. These studies have resulted in rich resources leading to critical insights into intricate biological regulatory systems. However, the large number of novel PPIs discovered in these large-scale studies demonstrate that only a small portion of the PPI landscape is currently known[4,7–10]. Further, for specific diseases such as cancer, utilization of these large-scale datasets is limited by lack of inclusion of many disease-specific genes, lack of experimental data in relevant cellular environments as well as inconsistent PPI data quality among various databases. The human interactome space, particularly for disease-focused studies, remains largely open[4,5,11].

In contrast, cancer genomics studies have advanced towards comprehensive molecular characterization of human cancers, revealing an expanded cancer gene landscape[12,13] and defining a subset of the proteome that is intimately associated with cancer. Importantly, a large fraction of this cancer genomics space is occupied by non-enzymatic proteins that can only be therapeutically targeted through their molecular interactions[12,13]. To complement large-scale proteomics efforts and leverage cancer genomics data, we undertook a focused proteomics study to discover protein interactions among a set of genes selected for their involvement in lung cancer[14,15]. Our approach is supported by the understanding that proteins involved in a certain disease tend to interact with each other to form a disease-specific interaction network, such as those in cancer[16,17].

In this study, we establish a cancer-associated PPI network, termed the OncoPPi network (version 1), through implementation of a streamlined time-resolved Förster resonance energy transfer (TR-FRET) technology platform for systematic binary PPI discovery in an efficient high throughput format. We discover more than 260 high-confidence PPIs not identified in previous large-scale datasets. OncoPPi identifies prominent protein interaction hubs with new PPI partners revolving around key cancer drivers such as MYC, STK11, RASSF1 and CDK4, uncovers interactions for non-enzymatic proteins, suggests crosstalk between oncogenic pathways, implicates novel mechanisms of action for major oncogene drivers such as transcription factor MYC, and reveals connectivity of tumour suppressors with actionable targets, such as the highly altered lung cancer tumour suppressor STK11 with CDK4. The OncoPPi network expands the lung cancer-associated protein interaction landscape for discovery of novel cancer targets and connects tumour suppressors to available drugs, offering an experimental resource for exploitation of PPI-mediated cancer vulnerabilities.

## Results

**Defining the OncoPPi network.** To generate a cancer-focused PPI network, a robust PPI detection platform was established using TR-FRET technology to systematically map the association of a library of test proteins in a pairwise fashion[18]. A set of 83 genes was selected based on frequency of alterations in lung cancer and known involvement in cancer signalling pathways[19–21]. Genes are listed in Supplementary Data 1. Our miniaturized, TR-FRET-based PPI platform enables high throughput mapping in a mammalian cell environment. Due to the stringent proximity requirement ($<100$ Å) to obtain a positive FRET signal, the identified positive PPIs generally reflect direct interactions in protein complexes[18].

With streamlined workflow for PPI detection, we systematically tested the selected lung cancer gene set in a pairwise manner in H1299 lung cancer cells to characterize their inter-molecular connectivity. H1299 lung cancer cells provide a relevant cellular environment and, with high transfection efficiency, consistent TR-FRET assay performance. A total of 3,486 interactions were examined. To ensure a quality screening dataset, each PPI pair was tested with two fusion tags for each gene, triplicate samples and three independent rounds of screening with fusion vector-only negative controls plus positive controls and expression sensors included in parallel for each PPI pair, resulting in a total dataset of $>62,000$ data points (Fig. 1a). We defined a set of statistically significant PPIs (SS-PPI dataset, Supplementary Data 2) and a more stringent set of high confidence PPIs (HC-PPI dataset, Supplementary Data 2) based on statistical analysis of FRET signals (Fig. 1b). The SS-PPI set includes 798 interactions with $P \leq 0.05$. Through comparison to publically available PPI databases including the BioPlex human interactome[4], we identified 670 novel interactions and confirmed 128 previously reported PPIs as direct interactions in lung cancer cells (Fig. 1c, Supplementary Fig. 1). A set of 348 PPIs with stringent criteria (HC-PPI dataset, FOC $\geq 1.5$, q-values $\leq 0.01$) were selected and combined with experimentally confirmed PPIs to generate a set of 397 high confidence lung cancer-associated PPIs, forming the OncoPPi network (version 1) (Figs 1c and 2a, Supplementary Data 2). The experimental HC-PPI dataset is enriched for known PPIs (total 128), compared to the gene library as a whole ($P$-value $< 3.25 \times 10^{-11}$, hypergeometric distribution test). With 269 novel interactions, OncoPPi greatly expands the landscape of interactions among this selected set of lung cancer genes (Fig. 1d, Supplementary Fig. 2) and defines interactions with potential significance for cancer PPI target discovery, such as CDK4/STK11, LATS2/RASSF1 and MYC/NSD3 (WHSC1L1). These datasets are freely available through the NCI's Cancer Target Discovery and Development (CTD[2]) Network Data Portal (https://ocg.cancer.gov/programs/ctd2/data-portal) and Dashboard (http://ctd2-dashboard.nci.nih.gov/).

To assess the quality of our PPI datasets, we analysed data reproducibility and detection of false positive PPIs. We found that 97% (385) of 397 OncoPPi PPIs were detected in at least two independent experiments with fold-over control (FOC) $\geq 1.2$. Additionally, although PPIs are frequently detected in only one fusion orientation due to conformational restraints, among the 397 OncoPPI interactions, 53% (209 PPIs) were detected in both fusion directions with FOC $\geq 1.2$ (Supplementary Data 2). To evaluate the level of detection of false-positive PPIs, we overlapped our OncoPPi data with a reported set of ~2,000 non-interacting proteins (Negatome 2.0)[22]. Although, the Negatome shares only seven PPIs with the OncoPPi set, six of those PPIs were also negative in our screening (SFN/TSC1, YWHAZ/TSC1, TSC1/FOXO1, E2F1/SMAD2, SMAD2/RB1 and MAPK14/HRAS). One interaction reported in the Negatome as negative but positive in our screening was the AKT1/TSC1 PPI. However, AKT1 is a known regulator of TSC1/TSC2, and its specific interaction with TSC1 has been previously confirmed by co-immunoprecipitation from HEK293 cells[23]. We also examined our dataset for 14-3-3 protein interactions. Seven 14-3-3 isoforms are known to directly bind more than 200 different proteins through well-defined and highly conserved 14-3-3 binding motifs[24,25]. For the three 14-3-3 isoforms ($\sigma, \gamma, \zeta$) included in the screening, we detected well-validated interactors of 14-3-3 with known binding motifs, including RAF1, BRAF, ARAF, FOXO1, LATS2, YAP1, STK11 and PRAS40, as well as homo- and heterodimers of 14-3-3. Importantly, no 14-3-3 PPI was detected for a protein lacking a conserved 14-3-3 binding motif. These data suggest a high-specificity and low false-positive rate for PPIs in OncoPPi.

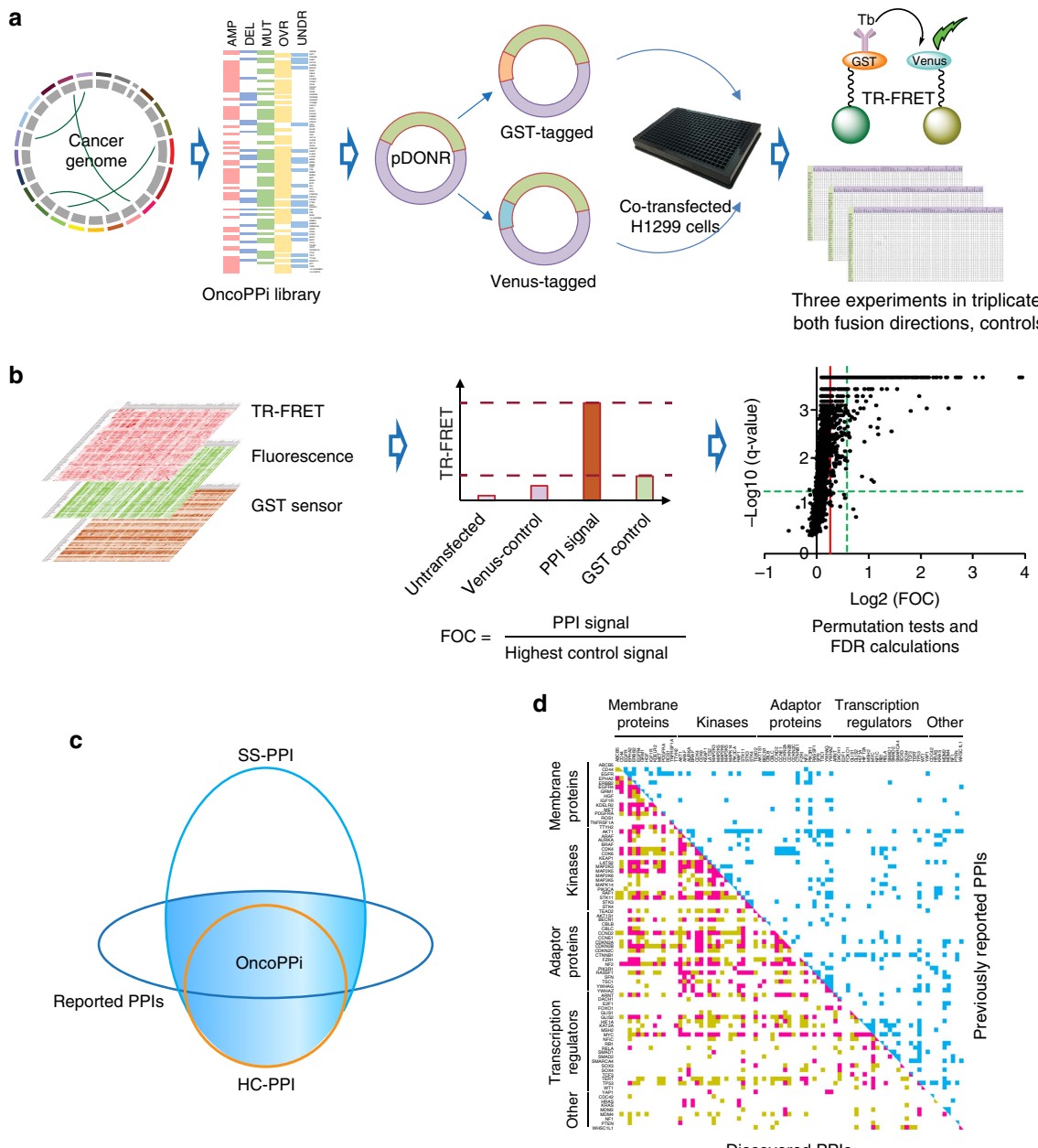

**Figure 1 | Design and workflow of the high throughput PPI screening platform leading to OncoPPi v1.** (**a**) Lung cancer genomics information was utilized to construct the OncoPPi expression vector library (see Supplementary Table 1) for pairwise TR-FRET-based high-throughput screening in H1299 lung cancer cells. (**b**) Analysis of high throughput PPI datasets included monitoring expression of each fusion protein construct using the fluorescence of the Venus-fusion protein and a FRET-based GST biosensor for the GST-fusion protein. Positive PPIs were defined based on FOC values, *P* values, and q-values calculated using the Benjamini–Hochberg procedure. See also Supplementary Table 2. (**c**) Definition of OncoPPi. Venn diagram representation of the OncoPPi network as a defined set of HC-PPIs plus previously reported interactions validated in these studies (Supplementary Fig. 2, Supplementary Table 2). (**d**) Schematic heatmap representation of OncoPPi expansion of the PPI landscape for lung cancer-associated genes, including membrane proteins, transcriptional regulators, adaptor proteins, kinases and others. Blue are previously described PPIs, magenta and yellow are experimentally determined OncoPPi and SS-PPI sets, respectively. See also Supplementary Fig. 2.

**OncoPPi network architecture reveals critical signalling hubs.** To gain structural insights into the OncoPPi network (Fig. 2a), we examined features of its network topology. Indeed, the OncoPPi network overall exhibits features of a scale-free network and matches the general characteristics of defined biological networks[16]. For example, the node degree distribution fits the Power law with a correlation coefficient of $R = 0.650$, while no correlation ($R = 0.367$) was observed for the distribution of node clustering coefficients. Comparing degree and betweenness centrality (BC)

index values for network nodes revealed MYC, AKT1 and STK11 as the three hub proteins most critical for the OncoPPi network (Fig. 2b), followed by RASSF1 and LATS2. On average, each protein in OncoPPi connects with nine protein partners, compared to an estimated median of five protein partners in the general proteome[4,5], supporting the notion that proteins involved in the same disease, such as cancer, tend to interact with each other[16]. Details on connectivity for each hub ($\geq 5$ partners) are presented in Figs 2 and 3 and Supplementary Table 1.

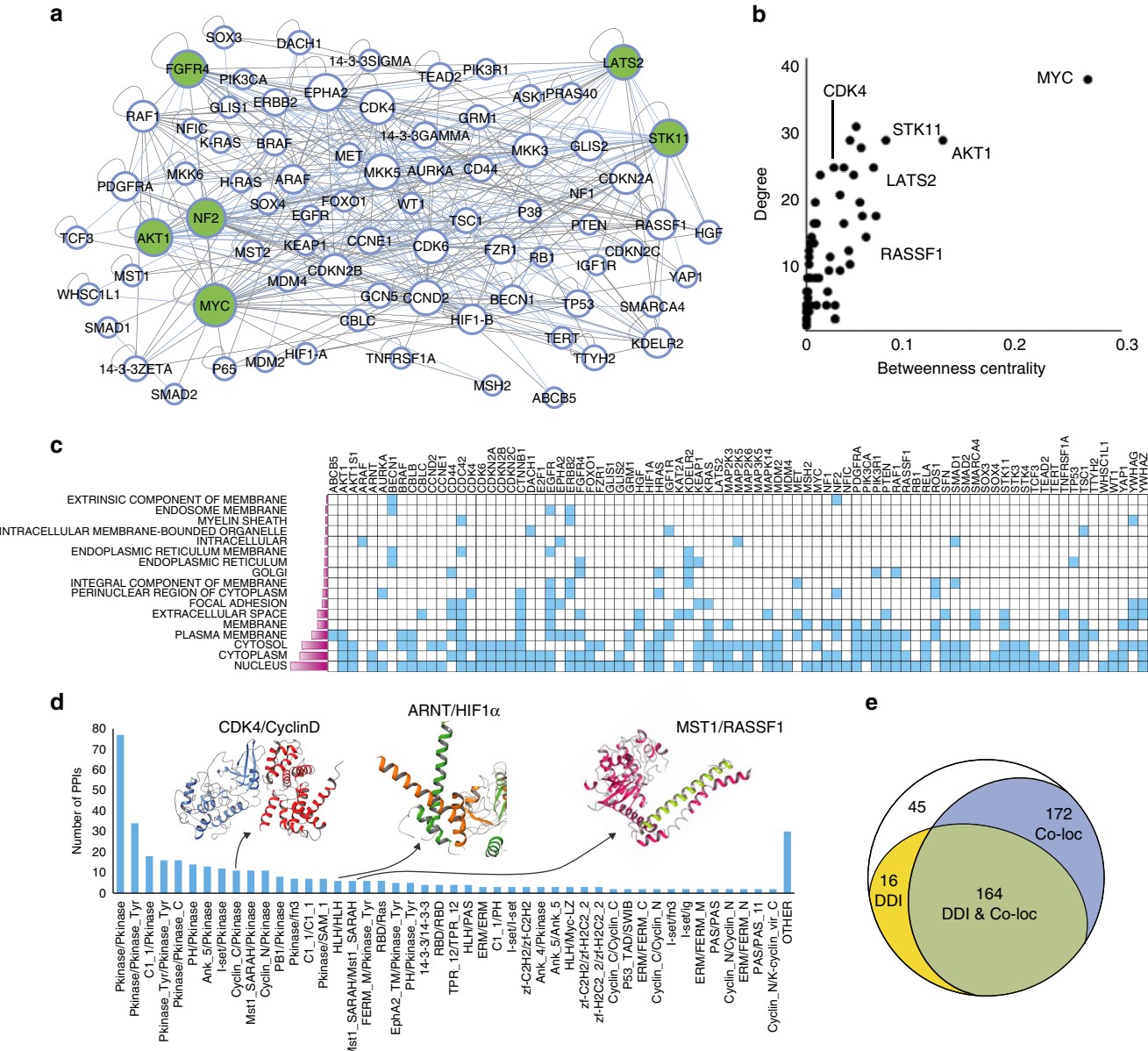

**Figure 2 | OncoPPi network architecture.** (**a**) A connectivity map of the OncoPPi network involving 83 lung cancer-associated proteins linked via 397 interactions (Supplementary Cytoscape file available for visualization and detailed analysis, see Supplementary Data 3). Major hubs highlighted in green. PPIs with mutual exclusivity of genomic alterations in LUAD are indicated with blue lines. (**b**) Analysis of network topology, including degree and BC reveals major PPI hubs. (**c**) Heatmap showing GO annotations of cellular localization for OncoPPi network genes. (**d**) Bar graph showing the number of OncoPPi PPIs supported by predicted domain–domain interactions. Domain–domain pairs in OncoPPi using Pfam domain annotation are listed on the x axis, the number of PPIs in OncoPPi on the y axis. Examples of co-crystallized CDK4/CyclinD, ARNT/HIF1α, and a homology model of MST1/RASSF1 constructed with the Swiss-Model server (swissmodel.expasy.org) based on MST1/RASSF5 crystal structure are shown to illustrate the interactions between different structural domains. (**e**) Venn diagram showing the distribution of PPIs in the OncoPPi network supported by cellular co-localization (Co-loc) data and/or structural domain–domain interactions (DDI). See also Supplementary Fig. 3.

To infer their functional association in cells, cellular co-localization and presence of shared interaction domains were assessed for each PPI pair. Cellular co-localization of each PPI pair was evaluated based on Gene Ontology (GO) annotations in the UniProt database (Supplementary Data 1). We found that 84% (336) of OncoPPi interactions share common GO cellular compartments (Fig. 2c, Supplementary Fig. 3). We next assessed presence of complementary and conserved structural domains and motifs that could mediate the observed PPIs. Pfam domains (Protein families database) were mapped to each OncoPPi protein (Supplementary Table 2) and integrated with known 3D domain–domain interaction structural data in the 3DID database[26] to

assign possible structural elements for mediating PPIs in the OncoPPi dataset (Fig. 2d, Supplementary Data 2, Supplementary Fig. 3). Forty-five percent (177) of the OncoPPi interactions shared potential complementary interaction domains compared to 16% for the overall gene set ($P$ value of $3.47 \times 10^{-49}$, Fisher's exact test, two-sided). Remaining PPIs in the network may utilize domains and motifs yet to be defined for their associations, offering opportunities for future discovery. Overall, 41% (164) of OncoPPi interactions share both co-localization and interaction domain annotations (Fig. 2e). Examples of proteins with shared domains and co-localization include CDK4/CCND2 (Cyclin/ protein kinase domains), ARNT/HIF1α (interaction between

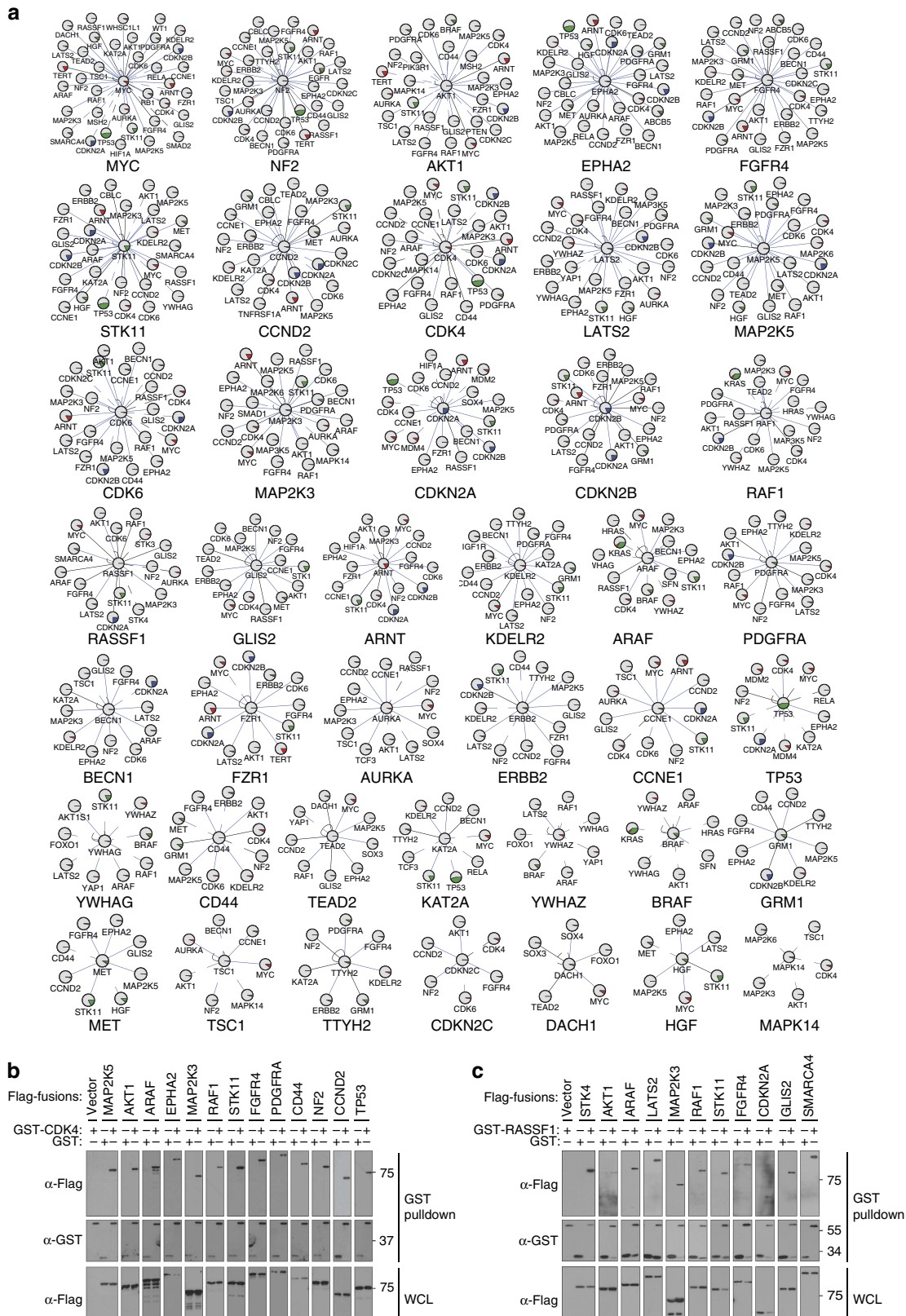

**Figure 3 | Major hub proteins in the OncoPPi network.** (**a**) Hub and spoke diagrams for the 41 proteins that form at least five heterodimers in the OncoPPi network are shown. The red, blue and green sectors inside the nodes represent the percent of LUAD cases (based on LUAD TCGA provisional dataset) with gene amplifications, deletions or mutations, respectively. PPIs identified as physical interactions in public IntAct, BioGrid, String or GeneMania databases are indicated with dashed lines (Supplementary Data 2). Newly discovered PPIs are indicated with solid lines. Functional connectivity between interacting partners was evaluated with mutual exclusivity analysis of genomic alterations (mutual exclusivity) and FUSION analysis (Supplementary Data 2). PPIs positive in mutual exclusivity or FUSION analyses are highlighted with blue lines. The proteins are identified by the Human Genome Organization Gene Nomenclature Committee approved symbols. (**b**) Confirmation of CDK4 and (**c**) RASSF1 PPIs with GST-pull down assay in HEK293T cells. Both Venus (vector) and GST tags were used as negative controls.

helix–loop–helix domains), and MST1/RASSF1 interacting through Salvador/Rassf/Hippo (SARAH) domains (Fig. 2d).

The OncoPPi network also revealed prominent PPI hubs with novel connectivity (Fig. 3a) exposing potentially critical biological insights for cancer genes such as *MYC*, *STK11*, *RASSF1* and *CDK4*. To assess experimentally the physical connectivity of PPIs in the OncoPPi network with an orthogonal approach, we selected two major hubs for confirmation with a conventional GST-pull down assay, the oncogene CDK4 and the tumour suppressor RASSF1 (Fig. 3b,c). For CDK4, of 15 novel PPIs tested, all CDK4 partners were positive in this solid phase-based pull-down assay except ARNT, LATS2 and GLIS2 (Fig. 3b; data not shown). Of note, bi-directional detection of PPIs did not strictly predict ability to confirm by GST pull down, as three confirmed CDK4 PPIs were detected in only one fusion orientation in the primary screen (RAF1, NF2, TP53). Similarly, 10 out of 13 RASSF1 binding partners were confirmed (Fig. 3c). Based on the high validation rate (>80%), the PPI hubs identified in our OncoPPi dataset provide a framework for PPI target discovery and network-based functional examination of oncogenic signalling (Fig. 3 and Supplementary Table 1).

**Integration of OncoPPi with cancer genomics data.** The OncoPPi network reveals a large number of potential PPIs for interrogation that suggest crosstalk between important cancer signalling pathways. We sought to leverage the available vast cancer genomics datasets to provide independent support for the involvement of the described PPIs in a common pathway. Computational approaches with genomic datasets, including mutual exclusivity and functional signature ontology (FUSION) analyses were utilized. Mutual exclusivity analysis takes advantage of the observation that alterations in genes participating in the same biological process tend not to occur together in the same cancer patient[27]. For mutual exclusivity analysis, the log odds ratio (OD) values were calculated for OncoPPi node alterations in TCGA lung cancer patient samples (Fig. 2a, Supplementary Data 1 and 2)[28]. Indeed, many known PPIs had positive mutual exclusivity scores (negative Log [OD]), including MDM2/TP53 and YAP1/TEAD. For the 397 OncoPPi interactions, we found that 257 PPIs showed potential mutual exclusivity in lung adenocarcinoma (LUAD) or lung squamous cell carcinoma (LUSC) TCGA samples, and 97 showed potential mutually exclusivity in both the LUAD and LUSC samples (Fig. 2a, Supplementary Data 2).

FUSION analysis was utilized as a separate orthogonal approach to evaluate functional connectivity of PPIs in OncoPPi v1 (ref. 29). As previously described, FUSION uses expression patterns for a defined endogenous reporter gene set (ACSL5, ALDOC, BNIP3, BNIP3L, LOXL2, NDRG1) to establish cellular response signatures to individual gene knockdowns. We hypothesized that functionally important OncoPPi interactions would have similar FUSION signatures for interacting protein pairs. FUSION reporter expression signatures were determined for each OncoPPi gene (Supplementary Table 3) and then a similarity matrix of reporter gene signatures using Pearson correlation values ($R$) as a distance metric (Supplementary Table 4) was constructed. We found that known PPIs involved in defined oncogenic pathways, including PI3K-AKT1-PRAS40-mTOR and CDK4-RB1-E2F1, had significant $R$ values ($|R| > 0.5$), providing validation of the FUSION approach for detecting functional connections (Supplementary Fig. 4, Supplementary Data 2, Supplementary Table 4). Overlaying the set of 397 OncoPPi interactions with FUSION positive PPI pairs revealed that 116 physical PPIs can be associated through FUSION ($P$ value of 0.0483 using hypergeometric distribution, data in Supplementary Data 2).

Positive mutual exclusivity and FUSION scores support pathway crosstalk mediated by interacting proteins. For novel interactions, these scores can be integrated with other functional data for PPI target prioritization and hypothesis testing. Thus, the OncoPPi dataset and the associated functional analysis are expected to inspire novel hypothesis-driven research for major cancer driver PPIs and their functional interconnections among critical signalling pathways. As examples, the oncogenic driver CDK4 showed not only expected interactions with known partners, cyclins and CDK inhibitors, but also several protein kinases (Fig. 3b). Growth factor receptors and mitogen activated protein kinases have well-established roles in regulating CDK4 activity primarily through transcriptional activation of cyclins and posttranslational modification of both cyclins and CDK4 inhibitors[30]. The physical interaction of CDK4 with the identified protein kinases may engage CDK4 at multiple stages of the cell growth regulatory pathways through membrane-associated receptors, PDGFRA and FGFR, and the MAPK signalling cascade, RAF1, MKK3 and MKK5, and stress-response kinases AKT1 and STK11. The directionality of these regulatory mechanisms remains to be established. RASSF1, a member of the Ras association domain family, is known as a tumour suppressor that inhibits RAF1 signalling and activates MST1[31]. The interaction of RASSF1 with the NF2-MST1/2-LAST2 module of the HIPPO pathway raises the possibility that RASSF1 may couple the membrane-associated NF2 complex to the core HIPPO signalling pathway. The interaction of RASSF1 with SMARCA4, on the other hand, may localize RASSF1 to the SMARCA4-mediated chromatin remodelling complex to regulate the expression of growth regulatory genes, such as CD44[32,33]. Mutual exclusivity and FUSION analysis also support placement of MYC in signalling pathways with transcription factors HIF1α/HIF1β (ARNT) and TEAD2 and an epigenetic modulator NSD3 (WHSC1L1)[34], suggesting a new mechanism for MYC regulation. These implicated pathways suggest novel models of cell growth control and potential PPI targets for further experimental exploration.

**OncoPPi network suggests a BRD4-NSD3-MYC pathway.** MYC, a global regulator of gene expression, is frequently amplified in a large number of tumours (10% of LUADs) and plays a prominent role in tumorigenesis. MYC contains multiple structural motifs that mediate interactions with a number of regulatory proteins[35]. Our OncoPPi network confirmed reported MYC binders, such as GCN5 and SMAD2, and also revealed 23 new potential interaction partners for MYC, including NSD3. To test an OncoPPi-generated hypothesis and gain mechanistic insights, the NSD3-MYC interaction was further examined.

NSD3 is a member of the nuclear receptor-binding SET domain (NSD) family of histone H3 lysine 36 (H3K36) methyltransferases. It is frequently amplified and functions as an oncoprotein in lung tumours and a range of other cancers[34]. NSD3 has two isoforms, a long form with the methyltransferase domain and a short form (NSD3-s) without the enzymatic activity. Interestingly, NSD3-s serves as an adaptor to link BRD4 to the CHD8 chromatin remodelling protein[36]. NSD3-s binding to MYC as revealed in OncoPPi may regulate MYC's access to designated chromatin complexes. Indeed, dose-dependent TR-FRET and affinity pull-down assay confirmed the interaction of NSD3-s with MYC (Fig. 4a,b). The endogenous NSD3-s/MYC complexes were also detected in lung cancer H1299 and H1944 cells under physiological conditions by co-immunoprecipitation (Fig. 4c,d). This MYC interaction appears to be mediated by a region C-terminal to the PWWP motif of NSD3-s (Fig. 4e,f). Supporting functional significance of the interaction, co-expression of NSD3-s, but not the MYC-binding defective

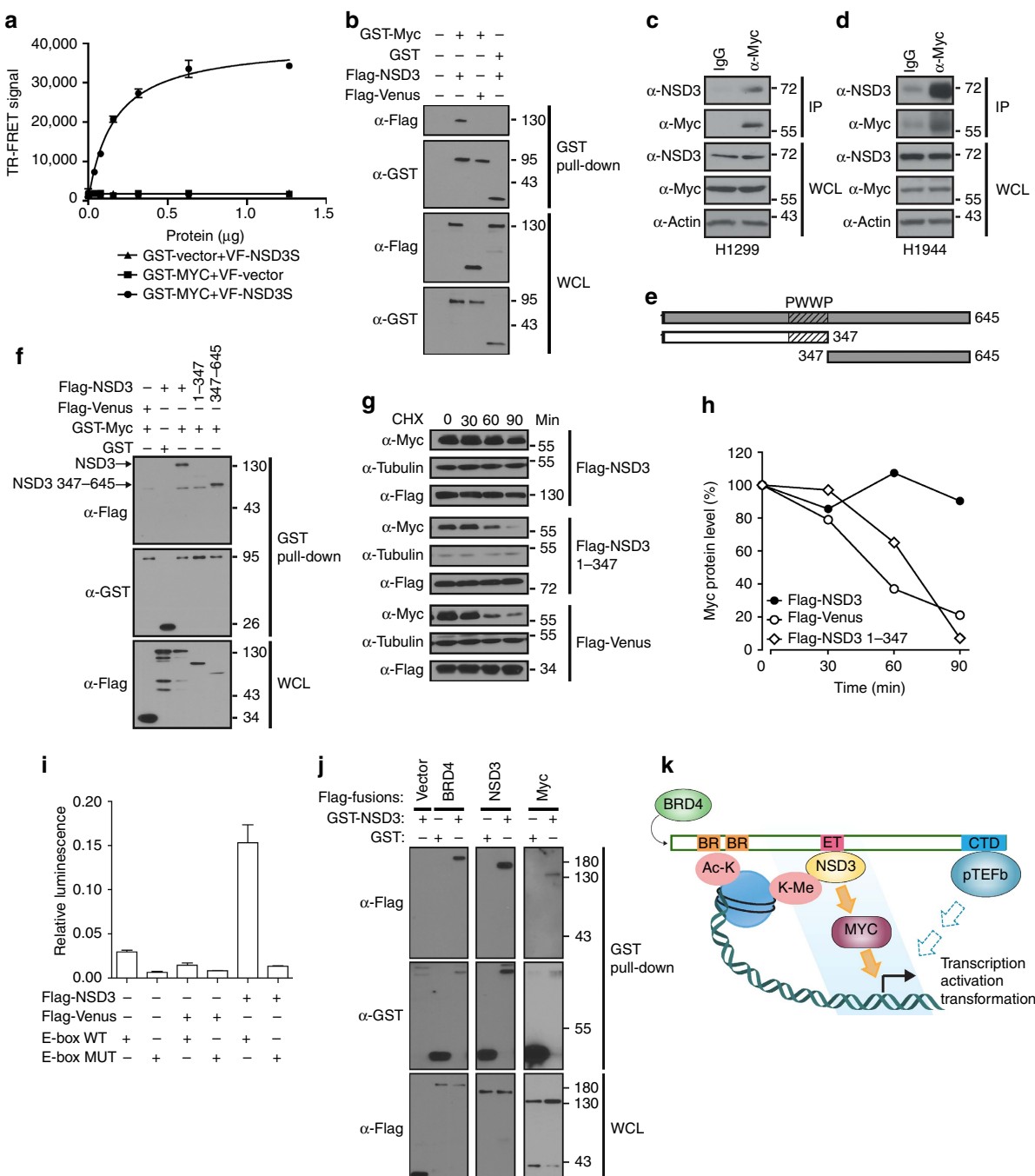

**Figure 4 | OncoPPi-suggested regulatory mechanism for the MYC oncogene.** (**a**) Direct interaction of NSD3-s with MYC. GST-MYC and Venus-Flag-tagged NSD3-s were expressed in HEK293T cells. Tb-conjugated anti-GST-antibodies were incubated with cell lysates to detect GST-MYC. The TR-FRET signal is expressed as the FRET ratio (520 nm/486 nm × 10⁴) Representative results of three independent experiments are shown. The error bars show the mean ± s.d. of three replicates. (**b**) NSD3–s/MYC interaction by GST-affinity pull-down assay. GST-MYC was captured by glutathione-resin to probe the presence of Flag-NSD3-s with western blotting. (**c,d**) Endogenous interaction of NSD3-s with MYC. The NSD3-s/MYC complex was co-immunoprecipitated with an anti-MYC antibody from lung cancer (**c**) H1299 and (**d**) H1944 cells with anti-IgG as control. (**e**) Schematic diagram of truncated NSD3 constructs. The MYC-binding fragments are indicated in grey. (**f**) GST-affinity pull-down assay with MYC and NSD3-s fragments. (**g**) NSD3-s stabilizes MYC. Immunoblot showing MYC and tubulin levels in HEK293T cells at different time points after inhibition of protein synthesis with cyclohexamide with or without co-expressed NSD3-s. (**h**) Graph of MYC protein levels at indicated time points based on densitometric analysis of results in (g). 100% corresponds to the total MYC detected at the 0 time point. MYC levels are normalized to tubulin protein levels. (**i**) NSD3-s activates MYC transcriptional activity. HEK293T cells were co-transfected with Venus-NSD3-s and either wild-type or mutant E-box luciferase reporter. Relative luciferase activity was measured, normalized to internal *Renilla* luciferase control. Representative results of three independent experiments are shown. The error bars show the mean ± s.d. of three replicates. (**j**) NSD3-s interacts with MYC, NSD3-s and BRD4. GST-NSD3-s was co-transfected with FLAG-tagged constructs for MYC, NSD3-s and BRD4 into HEK293T cells, followed by affinity chromatography with glutathione-conjugated beads, SDS-PAGE and immunoblotting with anti-Flag antibodies. (**k**) Proposed working model. BRD4 utilizes its ET domain to regulate MYC through a transcription-independent mechanism via the BRD4-NSD3-MYC pathway, in addition to the well-established BRD4-pTEFb-mediated pathway via the C-terminal fragment of BRD4. Both BRD4 and NSD3-s interact with modified histones.

fragment of NSD3-s (1–347), stabilized MYC protein (Fig. 4g,h) and increased MYC transcriptional activity as revealed by a MYC-driven reporter assay (Fig. 4i). These data led to the hypothesis that NSD3-s functions as a new activator of MYC oncogenic activity by bridging MYC with BRD4 to allow the regulation of MYC function in response to epigenetic modulators. Indeed, NSD3-s can be found in a complex with both MYC and BRD4 with distinct sites for respective interactions (Fig. 4e,f,j). Our results led to a working model that BRD4 regulates MYC through a transcription-independent mechanism by means of the BRD4-NSD3-MYC pathway, in addition to the well-established BRD4-pTEFb-mediated pathway (Fig. 4k), which may have significant clinical implications for the response of MYC-driven tumours to BRD4 inhibitors that are currently in clinical trials. Because both the short and the long forms of NSD3 can bind MYC (data not shown), it remains possible that both methytransferase-dependent and independent mechanisms are involved in the regulation of MYC by NSD3.

**Linking challenging targets with pharmacological agents.** Another important application of the OncoPPi network is to reveal potential intervention strategies for tumour suppressors through direct linkage to actionable cancer targets with FDA approved drugs. For example, major tumour suppressors, such as STK11 and TP53, could be connected with druggable targets like CDK4 and MDM2, respectively (Supplementary Fig. 5, Supplementary Table 5). STK11 is one of the most frequently mutated tumour suppressors in lung cancer (after TP53 and KRas)[37]. STK11 loss in an activated KRas mutant background significantly exacerbates tumorigenesis potential and tumour progression[38,39] supporting its critical role in maintaining normal cell growth. STK11 controls cell growth in part through its activation of the AMPK pathway[40]; however, the exact signalling pathways by which STK11 exerts its tumour suppressor function remain to be clarified. The interaction of STK11 with CDK4 leads to the hypothesis that the tumour suppressor function of STK11 may be in part mediated by engaging the cell cycle regulatory machinery through its direct interaction with and inhibition of CDK4 activity (Fig. 5a). Mutual exclusivity analysis with genomic alteration data showed that STK11 alterations in cancer patients rarely co-occur with CDK4, CDKN2A, CDKN2B or RB1, supporting a role for STK11 in an oncogenic pathway that is intimately associated with cell cycle function (Fig. 5b). Gene depletion phenotypes for STK11 and CDK4 appear to anti-correlate as shown by FUSION analysis (Fig. 5c). As such, STK11 may be physically and functionally associated with CDK4. This mechanistic relationship would nominate STK11 mutant lung cancer for therapeutic intervention with CDK4 inhibitors. Indeed, using STK11 copy number and mRNA expression data in the Cancer Cell Line Encyclopedia to define cell lines with high (STK11 High, 41 cell lines) and low (STK11 Low, 22 cell lines) STK11 expression, we observed significantly greater palbociclib sensitivity ($P = 0.002$, two-sided $T$-test) for the STK11 Low group (Supplementary Fig. 6, see Discussion).

To examine the potential connectivity of STK11 with CDK4, both physical and functional interaction of STK11 and CDK4 were examined. First, the interaction of STK11 with CDK4 in live cells was confirmed with a *Renilla* luciferase-based protein fragmentation complementation assay (Fig. 5d). Similarly, STK11 showed a significant signal with CCND2, a regulator of CDK4, supporting the interaction of STK11 with the CDK4/CCND2 complex. Endogenous co-immunoprecipitation experiments demonstrated the STK11/CDK4 interaction under physiological conditions in multiple lung cancer cell lines, including H1299 and H1792 cells (Fig. 5e,f). Further, knockdown

of STK11 was correlated with enhanced CDK4 kinase activity as shown by increased phosphorylation of RB (pRB), a physiological CDK4 substrate (Fig. 5g). Reconstitution of the knockdown cells with STK11 reversed the effect, suggesting a functional interaction.

To examine the therapeutic potential of the STK11/CDK4 connectivity, we utilized an FDA-approved small molecule CDK4/CDK6 inhibitor, palbociclib/PD-0332991, to probe the effect of CDK4 inhibition on STK11 activity[41]. Treatment of H1792 lung cancer cells (Fig. 5h) with palbociclib led to reduced pRB. Interestingly, reduced pRB triggered by palbociclib was correlated with reduced STK11 present in the CDK4 complex (Fig. 5i), suggesting release of STK11 from the CDK4 complex. It is unclear if the differences in concentration required for disruption of the CDK4/STK11 complex vs CDK4 catalytic inhibition are due to differences in sensitivities of techniques or reflect different biological mechanisms. It is possible that palbociclib could have a dual effect: not only inhibiting the CDK4-pRB axis, but also activating the STK11-pAMPK pathway. Indeed, treatment of lung cancer H1792 cells with palbociclib enhanced pAMPK and inhibited pRb (Fig. 5h). The pAMPK effect appeared to be STK11 dependent. Lung cancer A549 cells with defective STK11 (Q37*) (data not shown) or cell lines with STK11 knockdown showed undetectable effect on pAMPK when treated with palbociclib (Fig. 5h).

This observation led to the examination of the effect of STK11 genomic status on cell response to palbociclib. A pair of isogenic cell lines with differential STK11 gene status was utilized in a cell viability assay. It was shown previously that reduced STK11 was associated with increased sensitivity to phenformin[42], which was confirmed and used as a positive control (Fig. 5j). Similarly, when treated with palbociclib, STK11 silenced cells showed increased sensitivity over the parental H1792 cells (Fig. 5k). On the other hand, overexpression of the STK11 gene in STK11-negative H157 cells reversed their response to palbociclib, showing reduced sensitivity (Fig. 5l). Thus, loss of STK11 in lung cancer cells may lead to enhanced sensitivity to palbociclib, demonstrating an STK11-loss evoked enhanced-dependency of cells on CDK4. Due to extensive heterogeneity of NSCLC cell lines, a large panel of cell lines with multiple rigorous experimental approaches will be required to validate our observations. The use of two isogenic cell lines was the first attempt to establish any correlation between the STK11 status and sensitivity to palbociclib. The observed positive correlation supports future testing of this hypothesis to examine a potential therapeutic strategy for treating lung cancer patients harbouring STK11 alterations with a CDK4 inhibitor such as palbociclib.

## Discussion

This study reports the generation of an expanded lung cancer-associated PPI network, termed OncoPPi (v1), through the implementation of a TR-FRET-based high-throughput screening approach. The focused binary PPI screening coupled with a robust miniaturized screening platform allows a rigorous experimental design generating 18 data points for each PPI to ensure high-confidence PPI data for future hypothesis-driven investigations. OncoPPi unveils important PPIs in cancer not detected in previous interactome studies, and also integrates physical interaction, genomics and pharmacologic data to inform novel biology and therapeutic strategies. Previous large-scale interactome studies have indicated that much of the PPI landscape is still undescribed. Indeed, with our focused set of 83 lung cancer genes, we identify >260 novel interactions, expanding the PPI landscape for this gene set by >200%, including for well-studied cancer genes like MYC and CDK4.

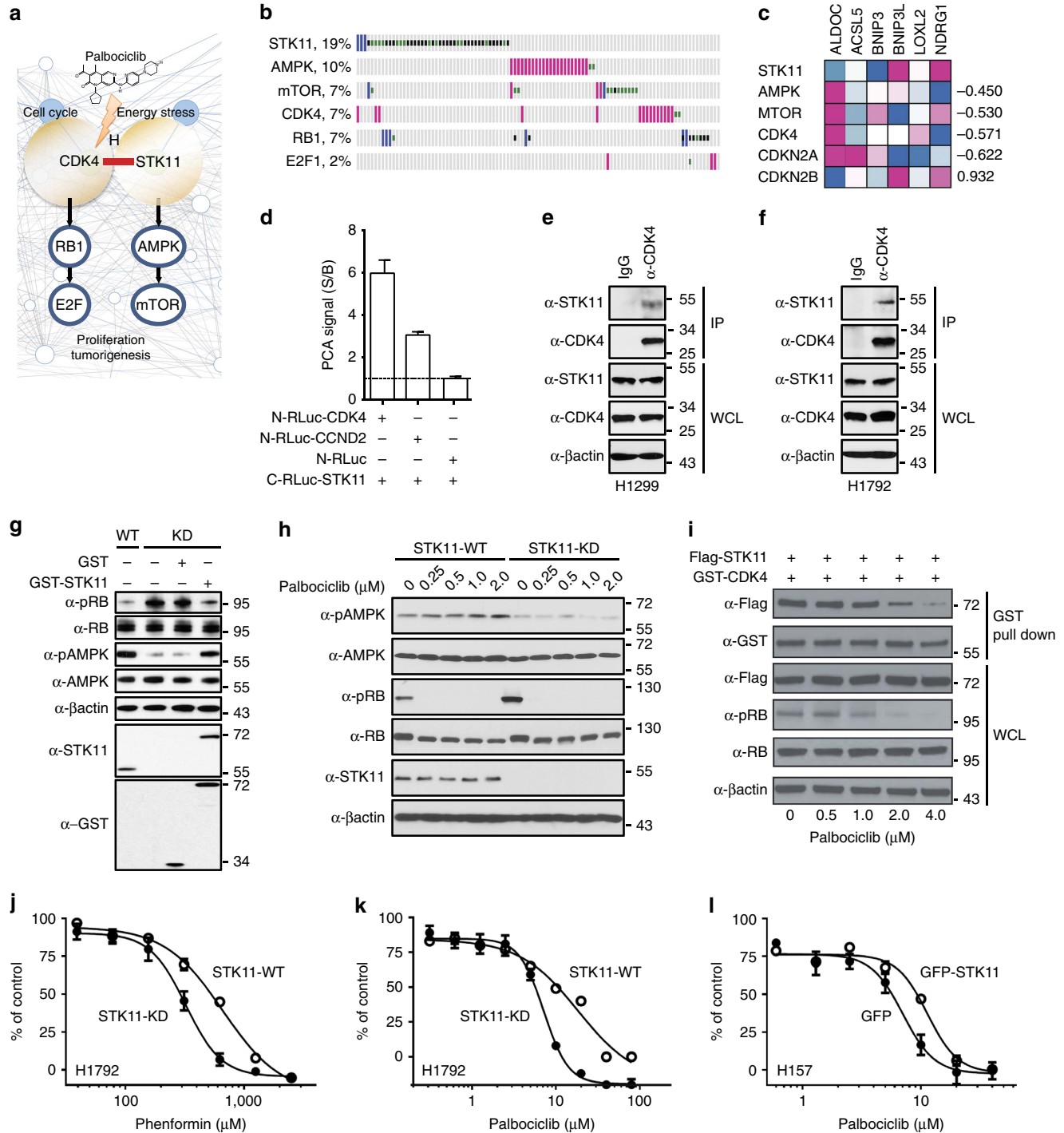

**Figure 5 | Physical and functional interaction of tumour suppressor STK11 with CDK4.** (**a**) The OncoPPi network links STK11 to CDK4 and palbociclib. (**b**) Mutual exclusivity map for the STK11 and CDK4 pathways in LUAD patient samples. Missense and truncation mutations are shown in green and black, respectively. DNA amplification is shown in magenta, deletions are shown in blue. Alterations of the PRKAA1 gene coding alpha 1 catalytic subunit of AMPK are shown. (**c**) Heatmap with expression signatures for FUSION reporter genes with individual knockdowns of STK11, AMPK, mTOR, CDK4, CDKN2A and CDKN2B based on Pearson correlation values from highest (blue) to lowest (magenta). (**d**) Interaction of STK11 with CDK4 and CCND2 using *Renilla* luciferase-PCA. Representative results of three independent experiments are shown. The error bars show the mean ± s.d. of three replicates. (**e**,**f**) Endogenous STK11 was co-immunoprecipitated with CDK4 in lung cancer (**e**) H1299 and (**f**) H1792 cells. (**g**) Effect of STK11 status on CDK4 and STK11 activity as shown by pRB and pAMPK levels in isogenic H1299 lung cancer cells (STK11 wild-type and knockdown). (**h**) Effect of CDK4 inhibition by palbociclib on pAMPK status in isogenic H1792 cells with wild-type (STK11-WT) and STK11 knockdown (STK11-KD). (**i**) CDK4 inhibition disrupts the interaction of CDK4 with STK11. HEK293T cells transfected with indicated plasmids were treated with palbociclib at the indicated concentrations. GST-CDK4 pull-down assay was carried out and protein expression was examined by western blot with the indicated antibodies. (**j**) Silencing of STK11 (H1792-408 cells) enhanced lung cancer cell response to phenformin and to (**k**) palbociclib in a cell viability assay. (**l**) Overexpression of STK11 in STK11 null cells (H157) reduced sensitivity of H157 cells to palbociclib. Western blots are representative of three independent experiments. Cell viability is expressed as % of control (mean ± s.d.).

Available to the community, OncoPPi is a valuable resource for discovery of cancer-associated PPIs as potential drug targets, examining network-informed signalling crosstalk, and predicting network-implicated tumour vulnerability, informative biomarkers and new strategies to target challenging cancer drivers.

Usefulness of high-throughput-derived PPI data depends on its reproducibility, as well as ability to detect true positive interactions. For the OncoPPi network, 97% of interactions were detected in at least two independent experiments (the other 3% were statistically significant, previously described PPIs). Fifty-three percent of OncoPPi interactions were detected in both fusion tag orientations. This result was expected, as structural factors can interfere with the ability to detect interactions for some orientations of fusion tags. Indeed, a number of previously known control PPIs (for example 14-3-3/YAP1 and CDK4/CDKN2A) and confirmed OncoPPi interactions (for example CDK4/RAF1 and CDK4/TP53, Fig. 3b and Supplementary Data 2) can be captured in only one of the fusion pairs. PPI results in Supplementary Data 2 include bi-directional detection information (column BFD) for further analysis by the scientific community. Overall, we found that >80% of tested OncoPPi interactions could be confirmed by GST pull down. Of note, unlike the TR-FRET assay, GST pull downs require stringent wash steps; thus, the 20% of PPIs not confirmed could represent both false positives and lower affinity interactions disrupted by wash steps. Our results with the Negatome and 14-3-3 binding motif indicate a low false positive rate for OncoPPi overall.

Despite transformative advances in cancer care towards precision oncology, clinical successes from genomics-based targeted cancer therapies remain largely focused on oncogenic enzymes, particularly protein kinases. Such enzymes offer druggable sites for therapeutic modulation of their catalytic activities. However, the majority of cancer driver genes identified through cancer genomics efforts encode 'undruggable' proteins such as (i) tumour suppressors with loss of function in cancer and (ii) adaptor proteins with no enzymatic activity, posing a major hurdle for therapeutic intervention[13]. These challenging cancer drivers function by participating in protein complexes that are involved in diverse cellular functions. Our approach addresses these challenges by placing cancer driver genes, both oncogenes and tumour suppressors, in the context of growth signalling PPI networks, offering unique opportunities to define promising therapeutic strategies for protein targets with or without enzymatic activity. The described PPIs that link adaptor proteins and tumour suppressors with well-annotated cancer drivers and actionable targets represent a wealth of potential new targets for therapeutic discovery and development (Figs 3–5). It should be noted that our TR-FRET-based high throughput PPI screening method utilizes co-expression of exogenous protein pairs. The presence of these interactions under physiological and pathological conditions should be validated to examine their functional importance. As examples, we illustrate OncoPPi-generated new hypotheses for future pathway perturbagen discovery through interrogating MYC and STK11 PPIs. Demonstration of two selected PPIs, MYC/NSD3 and STK11/CDK4, under endogenous conditions in multiple lung cancer cell lines strongly supports the validity of the OncoPPi network data set for further examination.

The MYC oncogene, which represents a highly validated and studied oncogenic cancer target, has no approved therapies that directly target the protein. This is due, in part, to the lack of enzymatic activity and defined catalytic site for structure activity guided design of potent compounds. Thus, it is recognized that the MYC interactome may represent a viable option to inhibit this pathway for therapeutic benefit[35]. Newly uncovered MYC

binding partners in the OncoPPi network serve as examples to illustrate OncoPPi-inferred new signalling pathways for regulation of MYC-driven cell growth and oncogenic programmes (Figs 2–4). We selected the physical interaction of MYC with NSD3 for validation, and uncovered a potential positive regulatory function for NSD3 in activating MYC, implicating the BRD4-NSD3-MYC pathway as a potential target for interrogating MYC-driven tumours. Because BRD4 inhibitors, such as JQ1, have been shown to be active against MYC-driven tumours, the intimate connection of NSD3-s with BRD4 and with MYC supports the hypothesis that NSD3-s may play a critical role in directing MYC-driven oncogenic programmes and may recruit MYC to a chromatin location through NSD3-s recognition of H3K36me3 and its association with acetylated lysine-BRD4, which warrants further investigation (Fig. 4k). The potential clinical importance of this connectivity is supported by the NSD3–MYC interaction in the NSD3-NUT fusion-driven NUT midline carcinoma[43]. It is very likely that the NSD3–NUT fusion may be critical for maintenance of MYC expression in these cancer cells. It is expected that other cancer drivers beyond MYC can also be exploited in a similar manner to guide functional validation and therapeutic discoveries.

Another application of the OncoPPi network is to reveal new tumour dependencies linked to tumour suppressor status (Fig. 5, Supplementary Fig. 5). Connectivity of STK11 with CDK4 suggests a potential application of CDK4 inhibitors for the treatment of patients with defective STK11. Here we demonstrated that STK11 directly interacts with and serves as a negative regulator of CDK4. Reduced STK11 levels have been correlated with enhanced CDK4 function. Thus, it is hypothesized that loss of STK11 in lung cancer may lead to enhanced dependency on upregulated CDK4. In support of this notion, lung cancer cells with reduced STK11 appear to be more sensitive than STK11 wild-type cells to palbociclib (Fig. 5). This observation is supported by the CDK4-mTORC1 pathway association as shown in breast cancer[44]. In addition, although heterogeneous mutational status of STK11 may be masked in large-scale genomics datasets, using Cancer Cell Line Encyclopedia datasets we observed increased palbociclib sensitivity for cell lines with low STK11 copy number and mRNA expression (Supplementary Fig. 6). Our results imply that it will be informative to use STK11 status as a biomarker for evaluating palbociclib efficacy. However, it should be cautioned that due to mutational heterogeneity of STK11 and the relatively high concentrations of palbociclib used to observe the drug response in lung cancer cells, future studies will be needed to probe the molecular basis for STK11-directed therapeutic strategies. We also note that palbociclib inhibits both CDK4 and CDK6, and we detected interaction of STK11 with both CDK4 and CDK6 (Fig. 3a). It is possible that the observed effect in Fig. 5 is due to dual inhibition of CDK4 and CDK6 by palbociclib, which requires further experimental examination. Similarly, other connectivity in the OncoPPi network may suggest therapeutic strategies for other actionable targets with approved drugs. Examples include dasatinib for EPHA2 and EPHA2-associated tumour suppressors, LATS2 and CDKN2B.

PPIs are highly promising targets for therapeutic discovery. This point is highlighted by recent clinical success in cancer immunotherapy. Current therapies re-direct immunological functions by interfering with the PD-1/PD-L1 interaction, supporting the concept of targeting PPIs in the immune system. To build on such success, our studies provide both potential PPI targets for manipulating oncogenic pathways, as well as highly sensitive high throughput screening (HTS) assays for PPI pathway perturbagen discovery. The TR-FRET assay format has been widely used in the HTS field for small molecule PPI modulator discovery[18]. Our results offer ready-to-go

TR-FRET-based HTS assays for both previously reported validated oncogenic PPIs, such as MDM2/TP53 and YAP/TEAD2, as well as for new PPIs with potential as therapeutic targets, such as MYC/NSD3-s and STK11/CDK4. These HTS assays are readily applicable for future HTS campaigns to discover PPI modulators.

## Methods

**Expression libraries for lung cancer-associated genes.** Genes collected for the current studies are listed in Supplementary Data 1 along with their sources, association with cancer (tumour suppressor/oncogene) and alterations in TCGA lung cancer datasets (provisional data sets downloaded 1/2016). Each gene was subcloned into the indicated Gateway entry vector (Invitrogen). The integrity of the genes was confirmed by BsrGI restriction digestion and by sequencing, generating the Entry-vector library. Genes in the entry vector library were transferred using the Gateway recombination system to destination expression vectors to produce a GST-gene fusion and a Venus-Flag-gene fusion for each gene, generating the OncoPPi expression vector library. PDONR223-EGFR, pDONR223-MET were gifts from William Hahn and David Root (Addgene plasmids #23935, #23889)[45]. pGBT9-NF1-GRD was a gift from Fuyuhiko Tamanoi (Addgene plasmid # 19993)[46].

**High throughput PPI screening.** We utilized the unique spectral overlap of terbium with Venus to develop a TR-FRET system that only requires the addition of one fluorophore during the assay process[47]. GST- and Venus-fusion proteins allow the coupling of anti-GST antibody-conjugated donor fluorophore, terbium, to fused Venus for FRET detection in solution to identify direct PPIs. For every PPI, GST- and Venus-only negative controls were included in parallel in every round of screening. To enable HTS for large-scale PPI detection, a cell lysate-based TR-FRET assay in a 384-well HTS format was developed. Briefly, H1299 lung cancer cells (2,500) were cultured in 384-well plates at 37 °C before they were co-transfected in wells with Venus-tagged genes in combination with GST-tagged genes using the Fugene HD reagent (Roche), assisted by robotic operations with the Sciclone ALH 3,000 liquid handling workstation (PerkinElmer). After incubation for 48 h, whole cell lysates were prepared by replacing the medium with lysis buffer (40 mM of Tris-HCl pH 8.0, 137 mM of NaCl, 1 mM of NaCl, 5 mM of NaF, 5 mM of NaPyrophosphate, 1% nonident P-40 (IGEPAL CA-630, Sigma-Aldrich) with proteinase inhibitors and phosphatase inhibitors) followed by a freeze-thaw cycle. Anti-GST-terbium antibody (Cisbio Bioassays Cat# 61GSTTLB, 1:1,000 dilution) was dispensed into each well with Multidrop Combi Reagent Dispenser (ThermoScientific). The lysate-antibody mixtures were incubated at 4 °C before the TR-FRET signal was recorded (EnVision reader setting: Ex 337 nm, Em1: 520 nm, Em2: 486 nm; mirror: D400/D505 dual; time delay: 50 μs). The TR-FRET signal is expressed as the FRET ratio (F520/F486 × 10[4]). The expression of test fusion genes was monitored through fused Venus or GST. The expression of Venus-tagged proteins was captured by fluorescence intensity (FI) (Ex 485 nm and Em 535 nm, mirror 505 nm). Venus-tagged gene expression was measured based on the Venus FI and calculated with the equation $FI = (FI_{signal}/FI_{background})$ where $FI_{signal}$ and $FI_{background}$ are the fluorescent signal of the cell lysates with Venus-tagged gene or untransfected cells, respectively. GST-tagged protein expression was monitored through a coupled GST-sensor assay (see below). With the established gene library and miniaturized assay, a pairwise TR-FRET screening was performed in triplicate and in both vector orientations (that is GST-Gene1/Venus-Flag-Gene 2 and GST-Gene2/Venus-FLAG-Gene 1), for a total of 3,486 protein–protein pairs × 2 orientations × 3 replicates × 3 independent experiments to generate 62,748 data points. To reduce false positive PPIs resulting from non-specific interactions between a 'bait' protein and the tag of a 'prey' protein (for example GST- or Venus-tag), each protein–protein pair (for example GST-protein 1/Venus-protein 2) was tested in triplicate in parallel with two corresponding matched negative control pairs (GST-protein 1/Venus-control, and GST-control/Venus-protein 2), with FOC calculated against the *highest matching control signal* for each individual PPI pair (see below).

**GST-fusion biosensor and experimental quality control.** To monitor the expression of test genes for the GST-fusion constructs, we developed a GST-biosensor for measuring GST-protein expression. The GST-biosensor is based on a TR-FRET assay, which includes two components, AlexaFluor555 conjugated-GST protein and anti-GST-Terbium (Tb) antibody. The purified GST-protein is labelled with AlexFluor555. The binding of anti-GST-Tb antibody to AlexaFluor555-GST protein brings TR-FRET donor (Tb) and acceptor (AlexFluo555) into proximity, leading to the generation of robust TR-FRET signal from GST-biosensor. Upon the addition of the GST-biosensor to PPI screening cell lysate, the expression of GST-tagged protein in cell lysate competes the binding of anti-GST-Tb to AlexFluor555, leading to the decreased TR-FRET signal. Therefore, the decreased GST-biosensor TR-FRET signal is correlated with the amount of GST-protein in the cell lysate.

**PPI network data analysis and statistics.** *Calculations of fold-over control values.* The FOC values were calculated for each PPI with the equation $FOC = Max\{[S_{G1V2}/Max(S_{G1},S_{V2})],[S_{G2V1}/Max(S_{G2},S_{V1})]\}$, where G1 is GST-tagged protein 1, V1 is Venus-tagged protein 1 and S is TR-FRET signal. For each experiment the average TR-FRET signals for the PPI ($S_{G1V2}$, $S_{G2V1}$), GST empty vector control ($S_{G1,G2}$) and Venus empty vector control ($S_{V1,V2}$) were calculated from the triplicates for each of two tested fusions (GST, Venus). Then, for each fusion the FOC values were calculated by dividing the average PPI signal by the maximum value of the two empty fusion vector controls. The maximum of the two FOC values obtained for the two fusions (GST, Venus) was considered as the final FOC value for a given PPI per experiment. Then, the average FOC values ($FOC_{AVR}$) were calculated by averaging the FOC values from individual experiments. These values were considered to be the final FOC value for a given PPI.

*Statistical significance.* The statistical significance of the differences in PPI and control signals was calculated with the rank sum permutation test. The calculations were performed using the MatLab package (MathWorks, Inc., Natick, MA, USA). The raw TR-FRET PPI signals detected for both fusions in triplicate in three independent experiments, and the corresponding signals of empty vector controls were used to build the PPI and Control groups, respectively. For each PPI the permutation test was performed according to the following procedure. First, for a given PPI pair, all PPI and control signals obtained in all experiments were ranked. The sum of the ranks of the PPI signals was calculated and was used as the test statistic for the permutation test. A total of 10,000 permutations were done. The P value was calculated with the equation $P = (N_s + 1)/10,001$, where $N_s$—number of cases where total ranks of shuffled labels exceed or are equal to that of true label. The false discovery rate and corresponding q-values were calculated with the Benjamini–Hochberg procedure; q-values less than 0.05 were considered significant.

*PPI mapping and network topology analysis.* The PPI network was constructed, visualized and analysed with Cytoscape 3.2 (ref. 48). Specifically, the node degree, BC index, network density and average number of neighbours were calculated using the Cytoscape Network Analysis tool. A tendency of nodes to form clusters can be characterized by cluster coefficients $C_i = 2n_i/k(k-1)$, where $n_i$—number of edges connecting $k_i$ neighbours of node i to each other.

*Defining the set of known PPIs.* Data for known PPIs were extracted from public PPI databases for the OncoPPi gene set (Supplementary Data 2). PPI databases utilized included: String[9], BioGrid[7], Intact[8], GeneMania[10] and BioPlex[4] and were limited to PPIs with reported physical association. For the OncoPPi gene set, a total of 364 PPIs were extracted from the databases, including 132 from String, 195 from IntAct, 260 from GeneMania and 281 from BioGrid. Overall, 73 PPIs were reported in all four databases, 90 in at least three databases, 105 in at least two databases and 96 PPIs appeared in only one of the four datasets.

*Co-localization and shared interacting domain analysis.* Protein localization data were extracted from the UniProt database based on GO annotations[49] for the set of OncoPPi genes (Supplementary Data 1). Analysis of co-localization was performed for the entire set of tested PPIs (Fig. 2c, Supplementary Data 2). Compartments with shared protein localization for OncoPPi PPIs included the extrinsic component of membrane, endosome membrane, myelin sheath, intracellular membrane-bounded organelle, intracellular, endoplasmic reticulum membrane, endoplasmic reticulum, Golgi, integral component of membrane, perinuclear region of cytoplasm, focal adhesion, extracellular space, membrane, plasma membrane, cytosol, cytoplasm and nucleus (Fig. 2c, Supplementary Data 2).

Structural domains for the OncoPPi genes were extracted from Pfam[50]. Data for domains involved in physical associations in co-crystallized proteins were extracted from the 3DID database[26] and utilized to annotate PPI pairs with known interacting domains for the 3,486 tested PPIs (Fig. 2d).

*Mutual exclusivity and FUSION analysis.* Mutual exclusivity analysis was performed in MatLab with the Cancer Genomics Data Server tool box[28,51]. The complete tumour samples from LUAD and LUSC TCGA Provisional datasets (downloaded January 2016, Supplementary Data 1) were used to analyse the mutual exclusivity of genomic alterations in lung cancer patient samples. Mutations, DNA amplifications and deletions were taken into account. Mutual exclusivity was evaluated in terms of log odds ratio (OD) values calculated as described previously[28]. The alterations of two genes were considered as mutually exclusive if Log(OD) < 0 (Supplementary Data 2).

Functional Signature Ontology (FUSION) analysis was performed as described[29]. Specifically, mRNA expression data of six reporter genes (ACSL5, BNIP3, BNIP3L, ALDOC, LOXL2 and NDRG1) obtained in HCT116 cells for individual gene knockdowns was used. Pearson correlations between mRNA expression patterns of the reporter genes corresponding to the knockdowns of two target genes were calculated with the MatLab package. The absolute value of 0.5 was used as a cut-off to determine functionally connected gene pairs. Correlations were obtained for all combinations of all genes tested in the HT PPI screening, except 14-3-3γ, GLIS1 and TTYH2, for which the corresponding data of reporter gene expression was not available (Supplementary Tables 3,4).

**Molecular biology techniques and cell culture conditions.** Standard molecular biology protocols were followed for making individual cloning vectors and truncation mutants. Human lung cancer cells H1299 (ATCC CRL-5803), H1944 (ATCC CRL-5907), H1792 (ATCC CRL-5895), H157 (ATCC CRL-5802), A549 (ATCC CCL-185) (ATCC, Manassas, VA) were cultured in RPMI 1640 containing

L-glutamine (CORNING Cat# 10-040) supplemented with 10% fetal bovine serum and 100 units/ml of penicillin/streptomycin. H157 lung cancer cells were used as STK11-null control for the expression of exogenous STK11 to generate isogenic STK11 cell line pair. Stable pLKO.1 vector control and LKB1-siRNA(408) H1299 and H1792 cells were created by lentiviral infection using specific siRNA constructs from Open Biosystems (Rockford, IL, USA) as described[52]. The stable transfected H157 (GFP) and H157 (GFP-STK11) cell lines were generated as previously described[53]. HEK293T (ATCC CRL-3216) cells (ATCC, Manassas, VA, USA) were maintained in Dulbecco's Modified Eagle's Medium (DMEM), with 4.5 g/L glucose, L-glutamine and sodium pyruvate (CAT# 10-013-CV CellGro) supplemented with 10% fetal bovine serum and 1X penicillin/streptomycin solution (CellGro). Cells were incubated at 37 °C in humidified conditions with 5% $CO_2$. For drug treatment for pathway analysis studies, cells were serum-starved for 24 h, then treated with either vehicle or the indicated concentration of the CDK4 inhibitor palbociclib (PD-0332991, SelleckChem) for 6 h. All cell lines have been tested for mycoplasma contamination.

**Affinity pull-down and co-immunoprecipitation assays.** For GST-affinity pull-down assays, cells were lysed in 1% NP-40 buffer (150 mM NaCl, 10 mM HEPES (pH 7.5), 1% nonident P-40 (IGEPAL CA-630, Sigma-Aldrich), 5 mM sodium pyrophosphate, 5 mM NaF, 2 mM sodium orthovanadate, 10 mg/L aprotinin, 10 mg/L leupeptin and 1 mM PMSF) and incubated with glutathione-conjugated beads (GE 17527901) for 2 h at 4 °C. Beads were washed three times with 1% NP-40 buffer and eluted by boiling in sodium dodecyl sulfate-polyacrylamide gel electrophoresis (SDS-PAGE) loading buffer. For immunoprecipitation, cell lysates were collected, quantified and were mixed with respective antibodies. For each co-immunoprecipitation, lysates containing ∼1.5 mg of total proteins were used and the antibody/lysate mixtures were incubated overnight at 4 °C. Then protein A/D agarose beads were added to the mixture followed by incubation at 4 °C for another 4 h. Beads were washed four times with lysis buffer, and proteins were eluted with SDS-PAGE sample buffer and analysed with indicated antibodies. The following primary antibodies were used for western blotting at the final dilution of 1:1,000 unless otherwise indicated: rabbit anti-GST (Santa Cruz Biotechnology, sc-459, 1:2,500 dilution); mouse anti-Flag (Sigma A8592, 1:2,500 dilution, or Sigma F3165, 1:2,500 dilution); rabbit anti-MYC (Santa Cruz, sc-764); mouse anti-RB (Cell Signaling, 9309 S); rabbit anti-phospho-RB (RB pS780, Cell Signaling 9307 S); rabbit anti-AMPK (Cell Signaling, 2532 L); rabbit anti-phospho-AMPK (Cell Signaling, 2535 L); mouse anti-β-Actin (Sigma, A5441); rabbit anti-LKB1/STK11(Cell Signaling, 3047 S); goat anti-NSD3 (Santa Cruz, sc-50152, 1:200 dilution). The mouse anti-CDK4 (Santa Cruz Biotechnology, sc-70831, 1:50 dilution) and rabbit anti-MYC (Santa Cruz, sc-764, 1:100 dilution) were used for immunoprecipitation assays. Uncropped gels for data presented can be found in Supplementary Figures 7–11.

**Protein stability assays.** Protein stability assays were performed according to established methods[54]. In brief, HEK293T cells were transfected using Xtreme-Gene (Roche 6366546001) following the manufacturer's instructions. After transfection, cells were incubated for 48 h in DMEM media supplemented with 10% FBS, then were treated with 100 µg/ml cycloheximide (2112, Cell Signaling) in DMEM media with 10% FBS. At the indicated times. 100 µl of 2X SDS-PAGE sample buffer was added and the cells were scraped from the wells, boiled for 5 min, then cell lysates were stored at − 80 °C. After all lysates were collected, each sample was loaded onto a 10% SDS-PAGE gel and then analysed by western blotting with rabbit anti-MYC antibody (5605, Cell Signaling, 1:1,000) to monitor MYC protein level. Protein expression was quantified from the western blot using GelQuant software. For analysis, MYC levels were normalized to tubulin protein levels (mouse, T-5326, Sigma, 1:2,000). Assays were performed three times.

**MYC reporter assay.** HEK293T cells were grown in six-well plates and transfected using Xtreme-Gene (Roche 6366546001) with Venus-FLAG-NSD3s or Venus vector along with Myc-Ebox-containing luciferase reporter plasmids, with either wild-type (GCCACGTGGCCACGTGGCCACGTGGC) or mutant (GCCTCGAGGCCTCGAGGCCTCGAGGC) E-boxes driving expression of firefly luciferase[55]. Renilla luciferase was included as an internal control. After transfection, cells were incubated for 48 h in DMEM media supplemented with 10% FBS. Cells were harvested mechanically, centrifuged at 1,600 r.p.m. for 10 min, and then re-suspended in 300 µl of DMEM media. The cells were transferred to 384-well plate, and the Myc reporter assay was performed using Dual-Glo luciferase kit (Promega, E2920) following the manufacturer's instructions. Firefly luciferase expression was normalized to the internal control Renilla expression. Data were analysed with Graphpad Prism software. Assays were performed three times.

**HTS cell viability assay.** Cells were seeded at 500 cells/well in 7 µl media in a 1536-well culture plate (Corning, Cat#3893) using a Multidrop Combi Reagent Dispenser (ThermoScientific) with the first column as a medium only control (blk). The next day test compounds (0.1 µl) were dispensed into wells in each plate using a Sciclone ALH 3,000 liquid handler (PerkinElmer) from a compound stock plate to give the indicated final concentrations. Each sample was tested with four replicates. After 3 days of incubation, Cell Titer Blue (1 µl; Promega, G8081) was

added to each well using the robotic liquid dispenser. The plate was incubated for 1–4 h at 37 °C. The FI of each well was read using an EnVision Multilabel plate reader (Ex 545 nm, Em 615 nm; PerkinElmer). % of control was calculated using the equation $(FI_{compound} − FI_{Avg.\ Blk})/(FI_{Avg.\ Neg} − FI_{Avg.\ Blk} \times 100)$.

**Data availability.** The data generated in this study are freely available in a public repository through the NCI CTD[2] Network (https://ctd2-dashboard.nci.nih.gov, https://ocg.cancer.gov/programs/ctd2/data-portal). The source and IDs of the genes and expression vectors used in the PPI screening are summarized in Supplementary Data 1. All experimental data are available from the authors.

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

## Acknowledgements

We thank Dr Kun-Liang Guan for the YAP1 vector, Dr Matthew Meyerson for the ARAF vector, Dr Wei Xu for the SMARCA4 vector, RIKEN Brain Science Institute for the Venus plasmid, Edward Prochownik for MYC reporter plasmids, and Fu lab members for helpful comments and experimental assistance. The results published here are in part based upon data generated by the TCGA Research Network: http://cancergenome.nih. gov/. This research was supported in part by NIH NCI U01CA168449 (HF), P01 CA116676 and 3P01 CA116676-05S1 (FRK), and a Georgia Cancer Coalition Award from Georgia Research Alliance (to HF, FRK), the Emory Chemical Biology Discovery Center, Winship Cancer Institute (NIH 5P30CA138292), and Emory University Research Committee grant (to A.I.). MAW and EAM were supported by R35CA197717 and U01CA176284.

## Author contributions

Z.l., A.A.I., R.S., V.G.-P., Q.Q., P.W., E.M., S.L., L.R., C.P., X.C. and X.-L.M. conducted the experiments and contributed to data interpretation; A.A.I., S.H., X.C., M.A.J., B.R., W.Z., A.M., M.A.W., C.S.M., L.A.D.C., Y.D., F.R.K. and H.F. participated in data analysis, discussion and manuscript preparation; Z.L., A.A.I., R.S., C.S.M., L.A.D.C., Y.D., F.R.K. and H.F. designed the experiments and wrote the paper; and all were involved in manuscript editing.

## Additional information

DOI: 10.1038/ncomms15350

# Corrigendum: The OncoPPi network of cancer-focused protein–protein interactions to inform biological insights and therapeutic strategies

Zenggang Li, Andrei A. Ivanov, Rina Su, Valentina Gonzalez-Pecchi, Qi Qi, Songlin Liu, Philip Webber, Elizabeth McMillan, Lauren Rusnak, Cau Pham, Xiaoqian Chen, Xiulei Mo, Brian Revennaugh, Wei Zhou, Adam Marcus, Sahar Harati, Xiang Chen, Margaret A. Johns, Michael A. White, Carlos S. Moreno, Lee A.D. Cooper, Yuhong Du, Fadlo R. Khuri & Haian Fu

*Nature Communications* 8:14356 doi: 10.1038/ncomms14356 (2017); Published 16 Feb 2017; Updated 18 Apr 2017

The original version of this Article contained an error in the spelling of the author Carlos S. Moreno, which was incorrectly given as Carlos Moreno. This has now been corrected in both the PDF and HTML versions of the Article.

