## [Peer review file · Nature Communications]

Peer Review File

Reviewers' comments:

Reviewer #1 (Remarks to the Author):

OncoPPI: A cancer focused protein-protein interaction network to inform biological insights and therapeutic strategies

In this manuscript, Zenggang and colleagues have developed protein-protein interaction networks in lung cancer cells to identify many new interactions. They have also reported new regulatory mechanisms for the genes such as MYC, STK11 and CDK4 in lung cancer. This manuscript also aims to associate tumor suppressors to drug targets and also explores therapeutic options. The presented dataset of protein-protein interactions in lung cancer cells would be useful for detailed mechanistic understanding of lung cancer related genes.

The high-throughput approach followed in the manuscript to identify PPI networks is impressive, but the conclusions drawn are not well supported from the reported experiments and need more evidence to strengthen them. In addition, many key methodological details are lacking, which make the manuscript hard to follow at many places in its current form. Some additional analyses would also improve the significance of this study. Specific comments that need to be addressed are listed below:

Major comments:

1. The high-throughput identification of PPI networks have historically had a high false positive rate. How false positive connections were identified and filtered out from the FRET experiments is not clear. The authors could consider a set of random pairs of genes, which are known to be non-binders as a “negative control” in their FRET experiments? Figure 2A shows very high connectivity of the genes altered in lung cancer, which doesn't seem plausible. It is not clear how each bait protein is interacting with almost every other protein considered in this study (Table S2). Is it because of the bias in the scoring scheme? Why FOC values were not normalized for signals of any two non-binders? In particular the choice of negative control is crucial for HT experiments such as these as they can be biased by promiscuity in interactors, and low off-rate of FRET signal. In addition, as a control the authors should show that FRET signal is independent of the single Venus or GST signal.
2. Authors have performed three independent experiments in triplicates for both fusions for 83 lung cancer genes in pairwise manner. However, they haven't discussed the reproducibility of interactions in different replicates. What is the specificity of the identified interactions? How correlated are signals when the Venus-GST pairs are swapped (so called A-B, B-A correlation)? These variables will help in identifying the high confidence interactors from such a huge datasets.
3. It is not clear if the overlap of the identified PPI dataset with other available datasets is significant. An enrichment analysis using known interactions can show the significance of the overlap. A sufficiently detailed venn diagram representation of overlap with all the known datasets (eg. STRING, Bioplex) can better explain how many known interactions were

captured.

4. Authors claim that STIK11 status can be used as a biomarker for evaluating the palbociclib efficacy. This hypothesis can be validated using the clinical datasets of lung cancer patients and cancer cell line encyclopedia (CCLE). This reviewer checked and observed no such trend.

5. It is known that BRD4 promotes transcription of MYC. Therefore, MYC driven tumors respond to BRD4 inhibitors. Although, it is interesting to know that NSD3 stabilizes MYC protein, how does a physical interaction of NSD3 with MYC and BRD4 add to this already known transcriptional mechanism action.

6. Overall the paper seems to focus on antiquated methods of network analysis, pioneered in yeast papers in early 2000's. For example, the scale free and betweenness nature, in this reviewer's opinion, offer no useful insight. Similarly, the conclusions in figure 5 are a stretch and are not supported by the data. I recommend its removal.

7. In Figure 6G the authors argue that STK11 knockdown regulates the activation of CDK4/Rb. I see no evidence for this in the figure. Also there appears to be a lack of consistency between panel b and f, as STK11 knockdown activates RB in panel f, but not in g. This discrepancy should be explained.

Minor comments;

7. Introduction lacks clear description of rationale and significance of why authors assume that the genes altered in lung cancer will interact among themselves. How exactly these genes were selected?

8. Heatmaps of the manuscript are unreadable and inappropriate to represent such PPI data. Authors need to explore more clear representations (Figure 1C, 2D)

9. Figure 2A: Map strength of interaction among these genes. It is not clear what authors wants to show in Figure 2b

10. It is exciting to see that authors could experimentally validate almost all the interactors of CDK4 and RASSF1 (Figure 3b). It will be useful to add a negative control panel (eg. Flag luciferase and GFP) to this pull-down blot.

11. The writing is poor at some places and should be improved.

12. What is the rationale behind using H1299 cells?

13. Authors have identified that 45% of OncoPPI interactions share known domain-domain interaction pairs. What is the p-value of this overlap?

14. Authors could associate 116 interactions through FUSION analysis. Is it statistically significant?

Reviewer #2 (Remarks to the Author):

Li et al. report on TR-FRET, a platform for systematically discovering protein-protein interactions. Using NCI-H1299 non-small cell lung cancer cells, they report on highly novel interactions for MYC, CDK4, STK11 and RASSF1. Further assessment of two such interactions, NSD3 and MYC, as well as STK11 and CDK4, are provided as validation of the methodology. Overall the manuscript is well written. Experiments validating the NSD3-MYC and STK11-CDK4 interactions could be strengthened.

Suggested improvements and data to add for revision:

1. Figure 4c. This is a critical panel demonstrating the interaction of endogenous NSD3 and MYC. The interaction appears weak. Was the reciprocal IP tried (i.e. IP with anti-NSD3 and Western blot with MYC). Additionally, it would be helpful to demonstrate this interaction in more than one cell line. Is BRD4 in the complex in these cells?
2. Figure 4d. The effect of NSD3 on MYC stability is shown via overexpression of NSD3. These results should be further validated with knockdown of NSD3 expression, with assessment of compromise of MYC stability. It is also important to demonstrate that the endogenous interaction shown in NCI-H1299 cells is critical for MYC stability in those cells.
3. Figure 4d and e. Has the interaction between NSD3 and MYC been mapped? The data set would be strengthened if it could be demonstrated that this interaction is essential for MYC stability.
4. Legend for Figure 4g should clarify which cells were transfected (presumably HEK293T).
5. The discussion should mention the identification of the NSD3-NUT fusion in NUT midline carcinoma that are very likely for maintenance of MYC expression in these cells (French et al. *Cancer Discov* 2014;4:928-41).
6. Figure 6e. The demonstration of endogenous CDK4-LKB1 complexes is limited to HEK293 cells; this should also be demonstrated in NCI-H1299 cells, and preferably in other NSCLC cell lines as well.
7. Figure 6g-l. The concentration of palbociclib used in these experiments is extremely high (2-100 μM). Typically, a maximum of 1 μM of this drug is used in vitro, with many cell lines demonstrating G1 arrest at 250 nM or less. If such high concentrations of palbociclib are to be used to demonstrate the biology, then the authors need to show that effects they are seeing are specific to CDK4. In panel g, is 2 μM really the lowest concentration required to block Rb phosphorylation? Similarly, in panel h, the disruption of the CDK4-LKB1 complex requires 4 μM palbociclib, which appears to produce an effect of less than a 2-fold decrease in complex formation. In panels k and l, the IC50 concentrations of palbociclib used in the more sensitive cell line of the isogenic pair are in the 5-10 μM range; again, these are extremely high concentrations of drug, raising questions about the degree to which viability is truly altered in the other cell line of the pair.
8. Data demonstrated in H1792 and H157 should be confirmed with the same cells used in

the original screen, namely H1299, as well as in cell lines that demonstrate anti-proliferative effects with concentrations of palbociclib more typically used (250 nM-1 μ M). Considering the substantial heterogeneity between NSCLC cell lines, it is important to validate this interaction in a variety of other lines.

9. In many solid tumor cell lines, the major D-type cyclins are cyclins D1 and D3. Any particular reason why cyclin D2 was chosen?

10. The authors should comment on CDK6, another major target of palbociclib. The interaction maps suggest interaction between CDK6 and LKB1; the possible effect of palbociclib on this interaction should also be explored.

Reviewer #3 (Remarks to the Author):

In this study, the authors generate a novel set of protein-protein interactions between cancer-associated proteins through high throughput time-resolved FRET. They use the results to generate a novel cancer-focused PPI network. They identify hub proteins, validate selected interactions, and finally suggest that their linking of “undruggable” targets to known drug targets might guide therapeutic targets. The manuscript is interesting, well-written, technically sound and of broad enough interest to motivate publication in Nat Com.

Comments on the method:

The interactions generated are between overexpressed proteins, which may not necessarily be relevant under endogenous conditions. This might be stressed more in the manuscript.

Comment 1.

Based on the experimental testing, the true positive rate appears to be high. It would be interesting to see an analysis of the false negative rate, I could not find such information.

Comment 2.

The connectivity map in Figure 2 a does not really add anything as it is impossible to distinguish any details. Major hubs are indicated to be green in the figure legend, but they appear to be yellow.

Comment 3.

Line 209. The statement reads “The interaction of RASSF1 with the NF2-MST1/2-LAST2 module of the Hippo pathway implies a scaffolding function of RASSF1 in linking the membrane associated NF2 complex to the core HIPPO pathway.”

This may be true but the authors do not present any evidence that the RASSF1-LATS2 and RASSF1-NF2 interactions can occur simultaneously. There is hence no evidence for scaffolding function, so I suggest the authors modify the statement. Also, it should be LATS2, not LAST2.

Response to reviewers' comments:

RE: *Li et al*, "The OncoPPI Network of Cancer-focused Protein-Protein Interactions to Inform Biological Insights and Therapeutic Strategies"

We thank the reviewers for their insightful and constructive comments on our manuscript. We appreciate that the reviewers found our "high throughput approach....to identify PPI networks is impressive", the manuscript is "interesting ... technically sound, and of broad enough interest", and "the presented dataset of protein-protein interactions ... would be useful for detailed mechanistic understanding of lung cancer related genes". However, to strengthen the current manuscript, Reviewer 1 would like to see "more evidence", additional "key methodological details", and "additional analyses". Reviewer 2 recommended that "experiments validating the NSD3-MYC and STK11-CDK4 interactions could be strengthened" while Reviewer 3 cautioned us about the "overexpressed proteins" and their relevance "under endogenous conditions". To address these concerns, as detailed below, we have revised the manuscript extensively including addition of new data and new analysis that strengthens our conclusions and we have added sections in the discussion about caveats of our detection system and its relevance to native states in cancer cells. We hope the reviewers agree that this revised manuscript is significantly improved and will serve as a useful resource for the research community.

Note: Line numbers indicated below refer to the submitted "redline" document in which line numbers are included and text edited in response to reviewer's comments is highlighted in red. To follow the guidelines of *Nature Communications*, the title has been revised to remove the punctuation.

Detailed Response to Reviewer's Comments

Reviewer #1:

We would like to thank Reviewer 1 for the detailed review and constructive criticism of our manuscript. We believe our high throughput approach focused on identification of positive PPIs among cancer genes has generated a valuable and novel dataset for the community. This reviewer's thoughtful questions have led us to significantly improve the manuscript, as addressed below.

Major comments:

1a. "The high-throughput identification of PPI networks have historically had a high false positive rate. How false positive connections were identified and filtered out from the FRET experiments is not clear. The authors could consider a set of random pairs of genes, which are known to be non-binders as a "negative control" in their FRET experiments?"

The reviewer's point about the identification and elimination of false positive hits is critical for generation of high-quality HTS data. To address this issue, we used several levels of quality control in our experimental design, such as (i) Venus- and GST-only vectors included as negative controls for every PPI tested; (ii) large numbers of replicates generated for both test and negative control PPIs to appropriately power the statistical analysis; (iii) all PPIs tested in both fusion orientations; (iv) multiple positive and negative controls included on every screening plate, and (v) rigorous statistical analyses performed to define significant signals (detailed below). With our stringent criteria, 90% of the total number of PPIs tested were negative and at least 80% of positive PPIs tested by GST pulldown were confirmed (see below). Further, our examination of a set of published "non-binding" negative controls were negative in our screen (see below). We have revised the text to clarify our use of negative controls in the text (Results, lines 112-3) and in the *Methods* section (lines 465-4, 516-5).

Statistical determination of positive PPIs: To reduce false positive PPIs resulting from non-specific interactions between a “bait” protein and a tag of the “prey” protein (e.g. GST- or Venus-tag), each protein-protein pair (e.g. GST-protein 1/Venus-protein 2) was tested in triplicate in parallel with two corresponding matched negative control pairs (GST-protein 1/Venus-control, and GST-control/Venus-protein 2), with fold-over control (FOC) calculated against the *highest matching control signal* for each individual PPI pair (Methods, lines 514-5). In addition to FOC cutoffs, statistical thresholds ($p < 0.05$) were applied to define a set of positive PPIs (Methods, lines 524-6). To identify PPIs with statistically significant TR-FRET signals, all triplicate raw TR-FRET signals generated in three independent experiments in both directions (18 data points for each PPI) and the signals of corresponding negative controls (36 data points) were subjected to a permutation test. The PPIs with $p < 0.05$ and at least 20% increase in TR-FRET signal over the highest matched negative controls (FOC) were defined as statistically significant PPIs (798), and the subset of 348 PPIs with at least 50% signal increase over the matched negative control signals and adjusted p-value (q-values) < 0.01 were defined as high-confidence PPIs. These high-confidence PPIs combined with previously reported statistically-significant PPIs were utilized to define the OncoPPI network of a total of 397 PPIs. With our rigorous statistical analysis 90% of the 3486 tested PPIs were called as negative.

Orthogonal confirmation of positive PPIs: To evaluate potential bias associated with the FRET-assay, an alternative GST pull-down assay was employed for validation of two selected large hubs of OncoPPI (CDK4 and RASSF1). Twelve of 15 novel CDK4 PPIs and 10 out of 13 novel RASSF1 PPIs detected in the HTS were also detected with the GST pull-down assay, suggesting that the PPIs included in OncoPPI network are not assay-specific, and at least 80% of OncoPPI PPIs can be detected in secondary assays (*page 8, lines 185-193* and Fig. 3). Of note, the GST pull-down assay uses cell lysates with stringent wash steps as compared to our homogeneous TR-FRET assay with no wash steps. Thus, the 20% of PPIs not confirmed with GST pulldown could represent false positives, or may also reflect differential sensitivity of TR-FRET vs GST pulldown for detection of PPIs.

“Non-binders” as a negative control: Identification of true non-binding proteins is as challenging as detection of true-positive PPIs. Recently, a “Negatome” set of ~2,000 PPIs tested but not detected in low-throughput experiments (e.g. endogenous co-immunoprecipitation) has been reported (Blohm et al., 2014). Although overlap of the Negatome set and PPIs tested in our screening is small (7 PPIs), six PPIs that were reported as negative in the Negatome were also negative in our screen. One interaction (AKT1/TSC1) reported as negative in Negatome was positive in our screening. This positive interaction is supported by a previously reported positive result from immunoprecipitation assays in HEK293 cells (Roux et al., 2004). These data are now included in Results (*page 6. Lines 135-142*).

14-3-3 as an example PPI: To address the specificity issue of the identified PPIs, we examined our dataset for the well-characterized adapter protein, 14-3-3, as an example. More than 200 binding partners have been reported for 14-3-3 and these binding proteins share a defined 14-3-3-binding motif that determines specificity of 14-3-3 PPIs (Fu et al., 2000; Johnson et al., 2010). Our OncoPPI gene set contains three 14-3-3 isoforms in our screening, 14-3-3 ζ , 14-3-3 σ , and 14-3-3 γ . Analysis of the OncoPPI network reveals that all of the binding partners detected for 14-3-3 ζ , 14-3-3 σ , and 14-3-3 γ have known 14-3-3 binding motifs and have been previously detected in various assays, including RAF1, ARAF, BRAF, FOXO1, LATS2, YAP1, STK11, and PRAS40. In contrast, no binding was detected between these 14-3-3 isoforms with proteins in the library lacking a 14-3-3 binding motif. These data are now included in the text (*page 7, lines 142-148*).

1b. *“Figure 2A shows very high connectivity of the genes altered in lung cancer, which doesn’t seem plausible. It is not clear how each bait protein is interacting with almost every other protein considered in this study (Table S2). Is it because of the bias in the scoring scheme?”*

Many of the proteins selected for our PPI screen are major regulators of oncogenic and apoptotic pathways and are known to interact with many protein partners involved in cancer, examples are 14-3-3 (see above) and MYC (Conacci-Sorrell et al., 2014; Tu et al., 2015). Previous studies have indicated that cancer-associated proteins tend to cluster together in PPI networks (Barabasi et al., 2011; Rolland et al., 2014; Vidal et al., 2011). Therefore, high-connectivity of major cancer drivers included in the OncoPPI network is in general agreement with expectations. Additionally, while Figure 2A may appear highly connected, only 397 PPIs out of 3486 tested (11%) are part of the OncoPPI network. See text in Discussion, lines 353-6 and also the Response above (1a) for our rigorous experimental approaches to address false positive issues.

1c. *“Why FOC values were not normalized for signals of any two non-binders?”*

For FOC calculations, rather than choosing two random non-binders, we chose a more conservative and stringent approach in which the TR-FRET signals of each PPI (GST-Protein 1/Venus-Protein 2) were normalized to the maximum value of the TR-FRET signals for the corresponding matched negative controls (GST/Venus-Protein 2 and GST-Protein 1/Venus). We clarified this point in the text (Results, lines 111-3) and the FOC calculations section of the Methods (lines 464-5, 514-5).

1d. *“In particular the choice of negative control is crucial for HT experiments such as these as they can be biased by promiscuity in interactors, and low off-rate of FRET signal.”*

As described above, both the Venus- and GST-vectors were included as negative controls for every PPI pair tested, we used the maximum value of the negative controls to stringently account for possible promiscuity in interactors, and a high percentage of positive PPIs could be confirmed with a more stringent, orthogonal GST pulldown assay.

1e. *“In addition, as a control the authors should show that FRET signal is independent of the single Venus or GST signal.”*

As described above, each protein-protein pair (e.g. GST-Protein 1/Venus-Protein 2) was tested in parallel with two corresponding matched negative control pairs (GST-Protein 1/Venus-control, and GST-control/Venus-Protein 2). Using the statistics described above, the criteria for scoring PPIs as positive requires the signal to be at least 50% greater than and statistically different ($p < 0.05$) from the matched negative controls.

2. *“Authors have performed three independent experiments in triplicates for both fusions for 83 lung cancer genes in pairwise manner. However, they haven’t discussed the reproducibility of interactions in different replicates. What is the specificity of the identified interactions? How correlated are signals when the Venus-GST pairs are swapped (so called A-B, B-A correlation)? These variables will help in identifying the high confidence interactors from such a huge datasets.”*

We apologize for this oversight and we have now included language describing the reproducibility of our data on page 6 (lines 131-2) and page 15 (lines 360-2). As described in the new text, our dataset was highly reproducible.

For the OncoPPI set of 397 PPIs, 97% (385 PPIs) were detected in at least two independent experiments with the FOC > 1.2, and 72% (98 PPIs) reproduced in all three experiments. For the 12 PPIs detected with FOC > 1.2 in only one experiment, the PPI signals (18 total datapoints) were statistically different from the negative controls with the permutation test. In addition, 8 out of these 12 PPIs (STK11 homodimer, RAF1 homodimer, TNFRSF1A homodimer, ARAF/14-3-3 sigma, TSC1/p38, AKT1/PTEN, RAF1/ASK1, and TCF3/GCN5) were supported by other experimental evidence as reported previously. Thus, we chose to include these 12 statistically significant PPIs in OncoPPI.

We have also added both data and text to the manuscript for analysis of swapped PPIs pairs (GST-Protein 1/Venus-Protein 2 vs GST-Protein 2/Venus-Protein 1). As described on *page 8 (lines 132-5) and page 15 (lines 363-9)*, 57% of PPIs in OncoPPI were detected in both directions. Due to structural factors that can interfere with the ability to detect interactions for certain orientations of fusion tags, it was expected that some protein interactions would be detected in only in one orientation. Indeed, a number of previously known control PPIs (e.g. 14-3-3 ζ /YAP1 and CDK4/CDKN2A), as well as some of the PPIs confirmed by GST pull down (ex. CDK4/RAF1) were detected in only orientation in the high throughput screen. This observation is consistent with our experience in the lab prior to this large-scale study and led to inclusion of both orientations in the experimental design. We have now incorporated the results for bi-directional detection of PPIs into Supplementary Table 2 to enable mining by the scientific community and we add discussion regarding configuration-dependent interactions on *page 15, lines 363-9*.

3. "It is not clear if the overlap of the identified PPI dataset with other available datasets is significant. An enrichment analysis using known interactions can show the significance of the overlap. A sufficiently detailed venn diagram representation of overlap with all the known datasets (eg. STRING, Bioplex) can better explain how many known interactions were captured."

This observation is important, both to validate our approach and to highlight the novelty of our dataset. We have computed the p-value for enrichment of known interactions detected in our set of high-confidence PPIs. Using the data in Supplementary Table 2, a total of 3,486 PPIs were tested, of which 364 were previously known in STRING, BIOGRID, INTACT, and/or GENEMANIA databases. Of the 348 high-confidence interactions detected, 79 were previously known, resulting in a p-value for overrepresentation of known PPIs of 3.25e-11 based on the hypergeometric distribution test. Thus, validating our approach, our network of detected high-confidence PPIs is significantly enriched with previously known PPIs. Further, we add 269 novel PPIs to the network for these important cancer genes. We have included the p-value for enrichment of known PPIs in the text on *page 6 (lines 122-4)*. The p-value for enrichment of known PPIs in the SS-PPI dataset was 5.13e-09, and for the OncoPPI dataset was 6.39e-39.

A new Supplementary Figure 1 with Venn diagrams showing the overlaps between each reference PPI database and the HC-PPI and SS-PPI datasets has also been added.

4. "Authors claim that STK11 status can be used as a biomarker for evaluating the palbociclib efficacy. This hypothesis can be validated using the clinical datasets of lung cancer patients and cancer cell line encyclopedia (CCLE). This reviewer checked and observed no such trend."

We appreciate the reviewer's astute observation regarding the ability to utilize existing large-scale genomics datasets (such as CCLE) for validation of our hypothesis. We would like to caution that STK11 status in large-scale datasets is complicated by complex genetic heterogeneity. However, based on the reviewer's comment, we performed a preliminary analysis using stringent STK11 expression cut-off (see below) and detected a potential association between STK11 expression status and palbociclib efficacy. However, given the caveats regarding STK11 status, we believe that this question will be most appropriately addressed with rigorous experimental studies. A cautionary note was included in the text

to address this important point along with a new Supplementary figure as stated below (*page 13, lines 299-05, page 14, lines 337-341, and page 17, lines 422-5*).

Text added to manuscript, page 13, lines 299-305: “This mechanistic relationship would nominate STK11 mutant lung cancer for therapeutic intervention with CDK4 inhibitors. Indeed, using STK11 copy number and mRNA expression data in the Cancer Cell Line Encyclopedia (CCLE) to define cell lines with high (STK11 High, 41 cell lines) and low (STK11 Low, 22 cell lines) STK11 expression, we observed significantly greater palbociclib sensitivity (p=0.002, two-sided T-test) for the STK11 Low group (Supplementary Fig. 6, see discussion)”.

Supplementary Figure 6 added: “STK11 expression correlates with response to CDK4 inhibitor palbociclib in the CCLE dataset.”

Text added to discussion, page 17, lines 422-5: “In addition, although heterogeneous mutational status of STK11 may be masked in large-scale genomics datasets, using CCLE datasets we observed increased palbociclib sensitivity for cell lines with low STK11 copy number and mRNA expression (Supplementary Fig. 6).

5. *“It is known that BRD4 promotes transcription of MYC. Therefore, MYC driven tumors respond to BRD4 inhibitors. Although, it is interesting to know that NSD3 stabilizes MYC protein, how does a physical interaction of NSD3 with MYC and BRD4 add to this already known transcriptional mechanism action.”*

The observed NSD3-MYC connectivity suggests a novel dual mechanism by which BRD4 activates MYC. It has been well established that BRD4 utilizes a transcription-dependent pathway via the BRD4-pTEFb interaction to engage the MYC super-enhancer. Our data suggest a transcription-independent pathway via the BRD4-NSD3-MYC (PPI) cascade. In support of this model, we have mapped the MYC binding site to the C-terminal domain of NSD3S (Fig. 4f), which is distinct from the BRD4 binding site at its N-terminal domain. This new mechanistic model has been described in the text (*page 12, lines 274-8*) and added as Fig. 4k. The heading for this section has been revised to emphasize the proposed new BRD4-NSD3-MYC pathway.

6. *“Overall the paper seems to focus on antiquated methods of network analysis, pioneered in yeast papers in early 2000’s. For example, the scale free and betweenness nature, in this reviewer’s opinion, offer no useful insight. Similarly, the conclusions in figure 5 are a stretch and are not supported by the data. I recommend its removal.”*

We appreciate the reviewer's comments about the methodologies utilized for network analysis. Our purpose of using these methods was to examine whether our OncoPPI network follows the general rules derived for biological networks, and to provide evidence for prioritization of hub proteins. These points could be made with a simple sentence in the text; accordingly, we have removed original Fig. 2b.

As a community resource, one application of the OncoPPI dataset is that it can be explored to identify new strategies for targeting tumor suppressors by placing them in the context of PPI networks and actionable oncogene drivers. As example, the STK11-CDK4 interaction was selected for detailed examination and revealed STK11 loss sensitized lung cancer cell lines to CDK4 inhibitor palbociclib. Our experimental evidence in support of the STK11-CDK4 demonstrates the value of such a concept illustrated in original Fig. 5 to generate new hypotheses for testing. Therefore, we opted to keep Fig. 5 but moved to the supplemental section as Supplementary Fig. 5.

7. “In Figure 6G the authors argue that STK11 knockdown regulates the activation of CDK4/Rb. I see no evidence for this in the figure. Also there appears to be a lack of consistency between panel b and f, as STK11 knockdown activates RB in panel f, but not in g. This discrepancy should be explained.”

Our previous panels were generated utilizing different treatment conditions. Therefore, we repeated these experiments with H1792 (original Fig. 6g) with similar conditions as in original panel f with H1299 cells. Under these experimental conditions, knockdown of STK11 in H1792 showed increased pRB (new Fig. 5h). The observed higher pRB and lower pAMPK level in H1792 (5h) than that in H1299 cells (5g) is likely due to amplified CDK4 in H1792 cells, which is consistent with our model of functional CDK4-STK11 interaction.

Minor comments:

7. “Introduction lacks clear description of rationale and significance of why authors assume that the genes altered in lung cancer will interact among themselves. How exactly these genes were selected?”

Large-scale interactome datasets indicate that only a small portion of the PPI landscape is known (Chatr-Aryamontri et al., 2015; Huttlin et al., 2015; Orchard et al., 2014; Szklarczyk et al., 2015; Warde-Farley et al., 2010). When we undertook the described studies, we hypothesized that this paucity of PPI data was also true for cancer gene networks. Instead of a large-scale interactome approach, we chose to perform a focused study on a relatively small number of selected cancer genes to allow use of multiple replicates, controls, and tag orientations, ensuring reproducibility and high quality data. Thus, we selected a set of genes involved in lung cancer based on frequency of alterations in lung cancer as reported in lung cancer genomics publications (Brambilla and Gazdar, 2009; Cancer Genome Atlas Research, 2014; Lawrence et al., 2014) and known involvement in cancer signaling pathways and, indeed, we found a large number of novel interactions even for well-known cancer genes (ex. *MYC*). Our results support the hypothesis that genes involved in the same disease, like cancer, tend to interact with each other to form a disease-network (Barabasi et al., 2011; Du et al., 2013; Rolland et al., 2014). Our goal for establishing the high quality OncoPPI network is to address the functional dimension of the cancer genome. We have revised the introduction (*lines 66-71*) to further clarify this rationale.

8. “Heatmaps of the manuscript are unreadable and inappropriate to represent such PPI data. Authors need to explore more clear representations (Figure 1C, 2D)”

Our intent with inclusion of the heat maps was to *schematically* illustrate how our experimental PPI dataset expands the PPI network for this set of lung cancer genes. To improve visualization and enhance the information content, we revised the heatmap in Figure 1 and removed the heatmap from Figure 2. For the heatmap in Fig. 1d, we now grouped proteins together based on protein types (ex. kinases, adaptor proteins), and we labeled the protein groups with a more readable font. We also include a larger version of the heatmap for readers to explore as Supplemental Fig. 2. For Fig. 2, we deleted the previous heatmap (was Fig. 2d) and created a larger version as Supplementary Fig. 3. To provide co-localization information, we have added a detailed (readable) heatmap as Fig. 2c. We have also added a bar graph showing the number of PPIs that share predicted interaction domains (Fig 2d). To show the relative numbers of PPIs in OncoPPI that share predicted domain-domain interactions and/or cellular co-localization we have added a Venn diagram (Fig 2e). OncoPPI PPIs supported by mutual exclusivity of genomic alterations are now highlighted by blue lines in the network Fig 2a. Also see #9 below for enhanced visualization.

9. *“Figure 2A: Map strength of interaction among these genes. It is not clear what authors want to show in Figure 2b.”*

The connectivity map in Figure 2A was intended for visualization and mining by readers. To achieve this goal for readers, we have now included a Cytoscape file for Fig 2a as a supplemental file which can be downloaded and used with the freely available Cytoscape program (<http://www.cytoscape.org>) for manipulation of the diagram. In the Cytoscape file, the strength of the interactions is reflected by line thickness. Because the quantitative results shown in original Fig. 2b are described in the text, Fig. 2b has now been removed.

10. *“It is exciting to see that authors could experimentally validate almost all the interactors of CDK4 and RASSF1 (Figure 3b). It will be useful to add a negative control panel (eg. Flag luciferase and GFP) to this pull-down blot.”*

In each of our GST-pull down experiments for Venus-tagged binding partners, we have used GST- and Venus-only expression vectors as controls. GST-control was included for each PPI pair in the original figure. For thoroughness as indicated by the reviewer, this Venus-vector control has been added to Fig. 3b,c for both CDK4 and RASSF1 hubs, as well as for Fig. 4j. Positive controls are also included, CCND2 for CDK4 and LATS2 for RASSF1.

11. *“The writing is poor at some places and should be improved.”*

We have worked to revise the text throughout the manuscript. We hope the reviewer finds the writing to be improved.

12. *“What is the rationale behind using H1299 cells?”*

Prior to our large-scale experiment, we explored the utility of a number of different lung cancer cell lines that enable HTS. Ultimately, we chose H1299 due to its relatively high transfection efficiency among lung cancer cells resulting in a robust and consistent performance for high throughput TR-FRET screening. H1299 is also a commonly used non-small cell lung cancer cell line with a defined genomic background that is readily available from ATCC with authentication information for quality control. We added a sentence to the text addressing this point (*page 5, lines 108-9*).

13. *“Authors have identified that 45% of OncoPPI interactions share known domain-domain interaction pairs. What is the p-value of this overlap?”*

We have computed the p-value of this overlap using Fisher’s exact test. The p-value for this test was 3.47e-49. This data is now provided on *page 8, line 174*.

14. *“Authors could associate 116 interactions through FUSION analysis. Is it statistically significant?”*

This result is marginally significant using the hypergeometric distribution at $p = 0.0483$. The p-value is now provided on *page 10, lines 222-3*.

Reviewer #2:

We thank Reviewer #2 for their overall assessment that we “report on highly novel interactions for MYC, CDK4, STK11 and RASSF1” and that “overall the manuscript is well written.” We have addressed Reviewer

#2's comments below, including addition of new data validating the NSD3-MYC and STK11-CDK4 interactions under endogenous conditions.

2-1. *"Figure 4c. This is a critical panel demonstrating the interaction of endogenous NSD3 and MYC. The interaction appears weak. Was the reciprocal IP tried (i.e. IP with anti-NSD3 and Western blot with MYC). Additionally, it would be helpful to demonstrate this interaction in more than one cell line. Is BRD4 in the complex in these cells?"*

We have carried out additional experiments and demonstrated endogenous NSD3-MYC interaction in additional lung cancer cell lines H1299 and H1944. New data have been included as *Fig. 4c* and *Fig. 4d*. We have tried and tested the commercially available NSD3 antibodies for reciprocal interaction and been unable to immunoprecipitate endogenous NSD3 so far, which is probably due to the poor quality of these antibodies for immunoprecipitation purpose. The presence of BRD4 in the NSD3 complex was detected under co-expression conditions, which is supported by previous publications (Shen, Ipsaro et al. 2015). Future experiments will address the detailed mechanisms of the BRD4/NSD3/MYC signaling complex under endogenous conditions. A new working model is proposed as *Fig. 4k*.

2-2. *"Figure 4d. The effect of NSD3 on MYC stability is shown via overexpression of NSD3. These results should be further validated with knockdown of NSD3 expression, with assessment of compromise of MYC stability. It is also important to demonstrate that the endogenous interaction shown in NCI-H1299 cells is critical for MYC stability in those cells."*

The knockdown of NSD3 experiments with RNAi have been reported by others, which showed reduced MYC levels in cells with NSD3 silencing, which supports our model (Shen et al., 2015).

To demonstrate the effect of endogenous *interaction* on MYC stability requires development of new reagents such as NSD3-MYC peptide disruptors or small molecule inhibitors, which is in progress. Toward this goal, we have mapped the MYC binding site to the C-terminal fragment of NSD3. We showed that NSD3s stabilized MYC; however, a MYC-binding defective fragment of NSD3s (1-347) was incapable of this function, supporting a specific role of NSD3 in MYC stabilization (*Fig. 4e-h*). Further refinement of the interface, detailed structural analysis and functional investigations will be carried out. However, we believe that these important studies are beyond the scope of the current OncoPPI network-focused manuscript.

2-3. *"Figure 4d and e. Has the interaction between NSD3 and MYC been mapped? The data set would be strengthened if it could be demonstrated that this interaction is essential for MYC stability."*

See Response to #2 above. We have mapped the MYC interaction site to the C-terminal fragment of NSD3. The use of the deletion NSD3 without the MYC binding domain failed to stabilize MYC. These results have been incorporated into the text and shown as *Fig. 4e-h*. Based on our results, future studies will develop NSD3-MYC PPI disruptors for examining MYC stability.

2-4. *"Legend for Figure 4g should clarify which cells were transfected (presumably HEK293T)."*

The legend has been revised to include the cell line: HEK293T.

2-5. *"The discussion should mention the identification of the NSD3-NUT fusion in NUT midline carcinoma that are very likely for maintenance of MYC expression in these cells (French et al. Cancer Discov 2014;4:928-41)."*

This clinically important point has been added to the discussion with the appropriate citation (*page 17, lines 409-11; (French et al., 2014)*).

2-6. *“Figure 6e. The demonstration of endogenous CDK4-LKB1 complexes is limited to HEK293 cells; this should also be demonstrated in NCI-H1299 cells, and preferably in other NSCLC cell lines as well.”*

We have examined endogenous interactions of CDK4 with STK11 using co-IP experiments in additional lung cancer cell lines, H1299 and H1792. Results were added *as Fig. 5e and 5f*.

2-7. *“Figure 6g-l. The concentration of palbociclib used in these experiments is extremely high (2-100 μ M). Typically, a maximum of 1 μ M of this drug is used in vitro, with many cell lines demonstrating G1 arrest at 250 nM or less. If such high concentrations of palbociclib are to be used to demonstrate the biology, then the authors need to show that effects they are seeing are specific to CDK4. In panel g, is 2 μ M really the lowest concentration required to block Rb phosphorylation? Similarly, in panel h, the disruption of the CDK4-LKB1 complex requires 4 μ M palbociclib, which appears to produce an effect of less than a 2-fold decrease in complex formation. In panels k and l, the IC50 concentrations of palbociclib used in the more sensitive cell line of the isogenic pair are in the 5-10 μ M range; again, these are extremely high concentrations of drug, raising questions about the degree to which viability is truly altered in the other cell line of the pair.”*

Our primary focus was to test any differential response of cells with varied STK11 status to palbociclib. Therefore, we first validated a positive control with the test isogenic cell lines for phenformin. Indeed, H1792 cells with STK11 silencing showed increased sensitivity to phenformin (Fig. 5j) (Shackelford et al., 2013). Using the same experimental design in parallel, the same isogenic H1792 cells with reduced STK11 showed increased sensitivity to palbociclib (Fig. 5k). In support of this correlation, expression of STK11 in STK11-defective H157 cells reversed the effect (Fig. 5l). Together, these observations are consistent with a STK11-dependent effect. This positive correlation is consistent with the OncoPPI-informed new hypothesis, which encourages future extensive experimental examination. The IC₅₀s for viability of lung cancer cells demonstrated in the current work are in general consistent with what has been reported in the literature for the specific lung cancer cell lines used, for example IC₅₀'s for H1792 and H157 are reported as 8 μ M (Barretina et al., 2012) and 9 μ M (Sumi et al., 2015), respectively. To be cautious, we have discussed the potential issues associated with relatively high concentrations of palbociclib used in the cell viability assay for lung cancer cells (*page 18, lines 426-32*).

The result in Fig. 5i was an unexpected observation. Palbociclib has a well-established role in inhibiting the catalytic activity of CDK4. Data in Fig. 5i showed that it also induces the dissociation of STK11 from the CDK4 complex. It is unclear if the differences in concentration required for disruption of the complex vs CDK4 catalytic inhibition are due to differences in sensitivities of techniques or reflect different biological mechanisms. We have added text on page 14 to address these points (line 321-3).

2-8. *“Data demonstrated in H1792 and H157 should be confirmed with the same cells used in the original screen, namely H1299, as well as in cell lines that demonstrate anti-proliferative effects with concentrations of palbociclib more typically used (250 nM-1 μ M). Considering the substantial heterogeneity between NSCLC cell lines, it is important to validate this interaction in a variety of other lines.”*

We agree with the reviewer that it is important to validate this interaction in a variety of other lung cancer cell lines. However, due to extensive heterogeneity of NSCLC cell lines, examination of functional

implications of this interaction requires a large panel of cell lines with multiple rigorous experimental considerations. One of such considerations is genetic background. The Rb1/CDK4 signaling can be disrupted in lung cancer cells by various mechanisms, such as RB1 deletion, CDK4 amplification or p16 deletion. In addition, recent studies indicated that KRas mutations may also affect cell sensitivity to CDK4 inhibitors (Puyol et al., 2010). We are developing an experimental strategy to comprehensively test the effect of this novel interaction in various genetic background in lung cancers [REDACTED]. The use of two isogenic cell lines was the first attempt to establish any correlation between the STK11 status and sensitivity to palbociclib. The observed positive correlation provides a foundation for future testing of this exciting hypothesis for translational research. This brief discussion has been added (page 14, lines 337-341; and page 18, lines 426-432).

2-9. *“In many solid tumor cell lines, the major D-type cyclins are cyclins D1 and D3. Any particular reason why cyclin D2 was chosen?”*

We aim to include all three D-type cyclins as we expand our OncoPPi library and screening for the second phase. In the first version of our OncoPPi library, we collected genes that are relevant to lung cancer and were readily available for cloning. Cyclin D2 levels have been associated with patient survival in non-small cell lung cancer (Ko et al., 2012) and we had a full-length clone available to include in the version 1 library.

2-10. *“The authors should comment on CDK6, another major target of palbociclib. The interaction maps suggest interaction between CDK6 and LKB1; the possible effect of palbociclib on this interaction should also be explored.”*

Based on the TCGA data, CDK4 is amplified or overexpressed in 16% of lung cancer adenocarcinoma samples, and CDK6 in 8% of samples. While we focus here on CDK4 – STK11 as an example, we did also detect interaction of CDK6 with STK11 in our screening and it is known that palbociclib also inhibits CDK6. The possibility of inhibiting both CDK4 and CDK6 by palbociclib is discussed in the text (*page 18, lines 430-2*). Our future work will explore the impact of palbociclib on CDK6 as well as the many novel interaction partners for STK11; however, we believe these studies are beyond the scope of this OncoPPi-focused manuscript.

Reviewer #3:

We appreciate Reviewer #3’s assessment that “The manuscript is interesting, well-written, technically sound and of broad enough interest to motivate publication in Nat Com.” We have addressed this reviewer’s comments as below, and we hope the reviewer agrees that the manuscript has been significantly strengthened and will prove to be a valuable resource for the readership of Nature Communications.

“Comments on the method: *The interactions generated are between overexpressed proteins, which may not necessarily be relevant under endogenous conditions. This might be stressed more in the manuscript.”*

We agree with the reviewer’s comment on one of the drawbacks of the method. Unfortunately, no methods currently exist for large scale systematic discovery of endogenous protein interactions in a binary high throughput screening format. We have included a section in the discussion that addresses the strengths and weaknesses of our method (*page 16, lines 388-90*). The validation of two selected PPIs, NSD3/MYC and STK11/CDK4, in multiple lung cancer cell lines under endogenous conditions supports our strategy.

“Comment 1. *Based on the experimental testing, the true positive rate appears to be high. It would be interesting to see an analysis of the false negative rate, I could not find such information.”*

False negatives could be estimated from the number of published PPIs that were not captured in our broadly defined positive PPI list as revealed by TR-FRET signals higher than the maximal signal from negative controls (FOC > 1). Among 364 published PPIs in our dataset, 325 were detected in our screening, and only 39 PPIs (~10%) were not, suggesting a relatively low rate of false negative PPIs. It is possible that some of these reported PPIs may be indirect, which could be missed from our stringent TR-FRET-based detection system. These data are included in Table S2 along with the FOC values, p-values, and q-values, for analysis by readers.

Also see Response to Reviewer 1, #1-#2 above.

“Comment 2. *The connectivity map in Figure 2 a does not really add anything as it is impossible to distinguish any details. Major hubs are indicated to be green in the figure legend, but they appear to be yellow.”*

We agree with the reviewer and apologize for the difficulties with reading of the network details. We have enlarged the network with slight improvement in visualization (Fig 2). Importantly, we have now also provided the network connectivity data as a Cytoscape file that allows readers to visualize and manipulate the network with the freely available Cytoscape program compatible with Windows, Mac OS, and Linux systems (<http://www.cytoscape.org/>). We thank the reviewer for pointing out possible confusion with the coloring of the major hubs. Now the standard RGB [0,255,0] green color is used to highlight the hubs.

“Comment 3. *Line 209. The statement reads “The interaction of RASSF1 with the NF2-MST1/2-LAST2 module of the Hippo pathway implies a scaffolding function of RASSF1 in linking the membrane associated NF2 complex to the core HIPPO pathway.” This may be true but the authors do not present any evidence that the RASSF1-LATS2 and RASSF1-NF2 interactions can occur simultaneously. There is hence no evidence for scaffolding function, so I suggest the authors modify the statement. Also, it should be LATS2, not LAST2.”*

We have modified the sentence and removed the “scaffolding” statement (see page 10, lines 238-41). We have also fixed the spelling of LATS2.

References

- Barabasi, A.L., Gulbahce, N., and Loscalzo, J. (2011). Network medicine: a network-based approach to human disease. *Nat Rev Genet* 12, 56-68.
- Barretina, J., Caponigro, G., Stransky, N., Venkatesan, K., Margolin, A.A., Kim, S., Wilson, C.J., Lehar, J., Kryukov, G.V., Sonkin, D., *et al.* (2012). The Cancer Cell Line Encyclopedia enables predictive modelling of anticancer drug sensitivity. *Nature* 483, 603-607.
- Blohm, P., Frishman, G., Smialowski, P., Goebels, F., Wachinger, B., Ruepp, A., and Frishman, D. (2014). Negatome 2.0: a database of non-interacting proteins derived by literature mining, manual annotation and protein structure analysis. *Nucleic acids research* 42, D396-400.
- Brambilla, E., and Gazdar, A. (2009). Pathogenesis of lung cancer signalling pathways: roadmap for therapies. *Eur Respir J* 33, 1485-1497.
- Cancer Genome Atlas Research, N. (2014). Comprehensive molecular profiling of lung adenocarcinoma. *Nature* 511, 543-550.

- Chatr-Aryamontri, A., Breitkreutz, B.J., Oughtred, R., Boucher, L., Heinicke, S., Chen, D., Stark, C., Breitkreutz, A., Kolas, N., O'Donnell, L., *et al.* (2015). The BioGRID interaction database: 2015 update. *Nucleic acids research* *43*, D470-478.
- Conacci-Sorrell, M., McFerrin, L., and Eisenman, R.N. (2014). An overview of MYC and its interactome. *Cold Spring Harb Perspect Med* *4*, a014357.
- Du, Y., Fu, R.W., Lou, B., Zhao, J., Qui, M., Khuri, F.R., and Fu, H. (2013). A time-resolved fluorescence resonance energy transfer assay for high-throughput screening of 14-3-3 protein-protein interaction inhibitors. *Assay and drug development technologies* *11*, 367-381.
- French, C.A., Rahman, S., Walsh, E.M., Kuhnle, S., Grayson, A.R., Lemieux, M.E., Grunfeld, N., Rubin, B.P., Antonescu, C.R., Zhang, S., *et al.* (2014). NSD3-NUT fusion oncoprotein in NUT midline carcinoma: implications for a novel oncogenic mechanism. *Cancer discovery* *4*, 928-941.
- Fu, H., Subramanian, R.R., and Masters, S.C. (2000). 14-3-3 proteins: structure, function, and regulation. *Annual review of pharmacology and toxicology* *40*, 617-647.
- Huttlin, E.L., Ting, L., Bruckner, R.J., Gebreab, F., Gygi, M.P., Szpyt, J., Tam, S., Zarraga, G., Colby, G., Baltier, K., *et al.* (2015). The BioPlex Network: A Systematic Exploration of the Human Interactome. *Cell* *162*, 425-440.
- Johnson, C., Crowther, S., Stafford, M.J., Campbell, D.G., Toth, R., and MacKintosh, C. (2010). Bioinformatic and experimental survey of 14-3-3-binding sites. *Biochem J* *427*, 69-78.
- Ko, E., Kim, Y., Park, S.E., Cho, E.Y., Han, J., Shim, Y.M., Park, J., and Kim, D.H. (2012). Reduced expression of cyclin D2 is associated with poor recurrence-free survival independent of cyclin D1 in stage III non-small cell lung cancer. *Lung Cancer* *77*, 401-406.
- Lawrence, M.S., Stojanov, P., Mermel, C.H., Robinson, J.T., Garraway, L.A., Golub, T.R., Meyerson, M., Gabriel, S.B., Lander, E.S., and Getz, G. (2014). Discovery and saturation analysis of cancer genes across 21 tumour types. *Nature* *505*, 495-501.
- Orchard, S., Ammari, M., Aranda, B., Breuza, L., Briganti, L., Broackes-Carter, F., Campbell, N.H., Chavali, G., Chen, C., del-Toro, N., *et al.* (2014). The MIntAct project--IntAct as a common curation platform for 11 molecular interaction databases. *Nucleic acids research* *42*, D358-363.
- Puyol, M., Martin, A., Dubus, P., Mulero, F., Pizcueta, P., Khan, G., Guerra, C., Santamaria, D., and Barbacid, M. (2010). A synthetic lethal interaction between K-Ras oncogenes and Cdk4 unveils a therapeutic strategy for non-small cell lung carcinoma. *Cancer Cell* *18*, 63-73.
- Rolland, T., Tasan, M., Charloteaux, B., Pevzner, S.J., Zhong, Q., Sahni, N., Yi, S., Lemmens, I., Fontanillo, C., Mosca, R., *et al.* (2014). A proteome-scale map of the human interactome network. *Cell* *159*, 1212-1226.
- Roux, P.P., Ballif, B.A., Anjum, R., Gygi, S.P., and Blenis, J. (2004). Tumor-promoting phorbol esters and activated Ras inactivate the tuberous sclerosis tumor suppressor complex via p90 ribosomal S6 kinase. *Proceedings of the National Academy of Sciences of the United States of America* *101*, 13489-13494.
- Shackelford, D.B., Abt, E., Gerken, L., Vasquez, D.S., Seki, A., Leblanc, M., Wei, L., Fishbein, M.C., Czernin, J., Mischel, P.S., *et al.* (2013). LKB1 inactivation dictates therapeutic response of non-small cell lung cancer to the metabolism drug phenformin. *Cancer Cell* *23*, 143-158.
- Shen, C., Ipsaro, J.J., Shi, J., Milazzo, J.P., Wang, E., Roe, J.S., Suzuki, Y., Pappin, D.J., Joshua-Tor, L., and Vakoc, C.R. (2015). NSD3-Short Is an Adaptor Protein that Couples BRD4 to the CHD8 Chromatin Remodeler. *Mol Cell* *60*, 847-859.
- Sumi, N.J., Kuenzi, B.M., Knezevic, C.E., Remsing Rix, L.L., and Rix, U. (2015). Chemoproteomics Reveals Novel Protein and Lipid Kinase Targets of Clinical CDK4/6 Inhibitors in Lung Cancer. *ACS Chem Biol* *10*, 2680-2686.

- Szklarczyk, D., Franceschini, A., Wyder, S., Forslund, K., Heller, D., Huerta-Cepas, J., Simonovic, M., Roth, A., Santos, A., Tsafou, K.P., *et al.* (2015). STRING v10: protein-protein interaction networks, integrated over the tree of life. *Nucleic acids research* 43, D447-452.
- Tu, W.B., Helander, S., Pilstal, R., Hickman, K.A., Lourenco, C., Jurisica, I., Raught, B., Wallner, B., Sunnerhagen, M., and Penn, L.Z. (2015). Myc and its interactors take shape. *Biochimica et biophysica acta* 1849, 469-483.
- Vidal, M., Cusick, M.E., and Barabasi, A.L. (2011). Interactome networks and human disease. *Cell* 144, 986-998.
- Warde-Farley, D., Donaldson, S.L., Comes, O., Zuberi, K., Badrawi, R., Chao, P., Franz, M., Grouios, C., Kazi, F., Lopes, C.T., *et al.* (2010). The GeneMANIA prediction server: biological network integration for gene prioritization and predicting gene function. *Nucleic acids research* 38, W214-220.

REVIEWERS' COMMENTS:

Reviewer #1 (Remarks to the Author):

The authors have addressed all significant concerns and the manuscript has been significantly improved.

Reviewer #2 (Remarks to the Author):

The authors have strengthened the data sets with additional experiments validating the NSD3-MYC and STK11-CDK4 interactions under endogenous conditions. Additionally, they have acknowledged concerns raised about the high concentrations of palbociclib used in their experiments in the revised discussion. Finally, they have responded to methodological and statistical concerns raised. The manuscript will catalyze multiple follow-up reports on the function and actionability of the protein-protein interactions defined and is suitable for publication in Nature Communications.

Reviewer #3 (Remarks to the Author):

Thanks for the revised version. To me the paper is ready to be accepted for publication.